# Ultra-fast switching memristors based on two-dimensional materials

S. S. Teja Nibhanupudi [1] ✉, Anupam Roy [1,2] ✉, Dmitry Veksler[3], Matthew Coupin [4], Kevin C. Matthews [4], Matthew Disiena[1], Ansh[1], Jatin V. Singh [1], Ioana R. Gearba-Dolocan[4], Jamie Warner [4], Jaydeep P. Kulkarni[1], Gennadi Bersuker[5] & Sanjay K. Banerjee[1] ✉

The ability to scale two-dimensional (2D) material thickness down to a single monolayer presents a promising opportunity to realize high-speed energy-efficient memristors. Here, we report an ultra-fast memristor fabricated using atomically thin sheets of 2D hexagonal Boron Nitride, exhibiting the shortest observed switching speed (120 ps) among 2D memristors and low switching energy (2pJ). Furthermore, we study the switching dynamics of these memristors using ultra-short (120ps-3ns) voltage pulses, a frequency range that is highly relevant in the context of modern complementary metal oxide semiconductor (CMOS) circuits. We employ statistical analysis of transient characteristics to gain insights into the memristor switching mechanism. Cycling endurance data confirms the ultra-fast switching capability of these memristors, making them attractive for next generation computing, storage, and Radio-Frequency (RF) circuit applications.

In today's digital era, the role of memory technology has become integral, particularly in the context of artificial intelligence (AI) applications. Several AI tasks, such as classification, recognition, natural language processing, prediction etc., heavily rely on large-scale memory storage and processing capability of the underlying hardware[1–3]. Conventional memory technologies like Static Random Access Memory (SRAM) or Dynamic Random Access Memory (DRAM) are energy-inefficient for implementing these data-intensive applications due to their volatile nature, resulting in dynamic as well as static power dissipation. These energy requirements are even more stringent for battery-operated mobile, wearable electronics, and Internet of Things (IoT) edge devices[4,5]. Although flash memory is non-volatile, the slow programming speed (>10us) makes it impractical for these applications[6,7]. To effectively harness and maximize the potential of AI and other emerging applications, an ultra-fast non-volatile memory is necessary.

Memristors are promising devices in the emerging non-volatile memory landscape that have shown great potential owing to their small footprint ($4F^2$), fast operation, and low power consumption[8–13].

Several studies explored the use case of memristors for storage, memory-centric computing[14–17], and high-frequency RF applications[18–20]. In the past decade, memristors fabricated using a variety of materials such as Transition Metal Oxides (TMO)[21–24], chalcogenides[25,26], and organic materials[27–29] have been extensively studied. TEM studies have conclusively established that the resistive switching in memristors can be attributed to the presence of conductive nanofilaments[30,31]. Based on the experimental observations, the switching dynamics in memristors have been captured using different modeling techniques[32–34]. The commercially available memristors typically switch in ~50 ns–100 µs range, making them slower than state-of-the-art DRAM[35–37]. Although a few studies have reported faster-switching memristors ($T_{SWITCH} < 5$ ns)[38–47], the fast-switching dynamics have not been thoroughly studied.

Here, we present an ultra-fast memristor fabricated using a two-dimensional (2D) material, i.e., hexagonal Boron Nitride (hBN) as the switching layer[48–50]. The atomically thin nature of the 2D material facilitates rapid filament dynamics enabling an ultra-fast and energy-efficient memristor. We study these memristors using ultra-short

[1]Microelectronics Research Center, The University of Texas at Austin, Austin, TX 78758, USA. [2]Birla Institute of Technology, Mesra, Ranchi 835215, India. [3]HRL Laboratories, Malibu, CA 90265, USA. [4]Texas Materials Institute, The University of Texas at Austin, Austin, TX 78712, USA. [5]M2D solutions, Austin, TX 78758, USA. ✉e-mail: subrahmanya_teja@utexas.edu; royanupam@bitmesra.ac.in; banerjee@ece.utexas.edu

pulses ($T_{PULSE}$ < 2.7 ns) in a custom-built RF test setup. Moreover, since the filament formation/dissolution in memristors is a stochastic process[51–53], we employ statistical analysis of the transient switching characteristics to study these memristors. Statistical data provided critical insights into the switching mechanism and the inherent stochasticity of the filament dynamics. To the best of our knowledge, this is the first study of this kind not just in 2D memristors but among the plethora of scientific reports based on TMO memristors. In addition, by leveraging cues from statistical data, we model the fast-switching dynamics using a combination of finite element-based physics solver, empirical, and analytical equations.

## Results

### Device fabrication and DC electrical performance

Structural features of the fabricated memristor with few-layer hBN sandwiched between active (Ti) and inert (Au) electrodes are shown in Fig. 1a, b. The switching hBN layers were synthesized using chemical vapor deposition (CVD) on copper foil and transferred onto Si/SiO2 substrate for device fabrication (see Supplementary Fig. 1 and Methods section for details). The final device structure was confirmed using scanning electron microscopy (SEM) (Fig. 1c) and cross-sectional transmission electron microscopy (TEM) (Fig. 1d) images. Synthesized hBN films are 7-8 layers thick with an inter-layer spacing of 0.33 nm, as evident from the TEM images (Supplementary Fig. 2,3). These CVD-grown films contain single atom and few atom-wide intrinsic defects, as seen in Fig. 1e. These defects facilitate the formation of conductive filaments, which, in turn, enable the desired resistive switching behavior. Notably, previous studies have reported the absence of a resistive switching phenomenon in exfoliated flakes that lack intrinsic defects[54,55]. The crystal structure and chemical composition of the synthesized hBN films were corroborated using Raman and X-ray photoelectron spectroscopy (XPS) techniques (Supplementary Fig. 4). In addition, memristors fabricated with thinner (3-4 layers) hBN films (Supplementary Fig. 5), synthesized by tuning the growth parameters were studied for comparison.

Devices with different cross-sectional areas varying from 500 × 500 nm² to 2 × 2 μm² were fabricated (Supplementary Fig. 6) and characterized. The majority of as-fabricated (virgin) devices (92%) exhibit high resistance (>1MΩ), which indicates the excellent quality of the synthesized hBN films (Supplementary Fig. 7.1). The remaining

devices with low initial resistance likely contain grain boundaries or multi-atom wide defects which were probably introduced during the fabrication processes. These leaky devices were discarded and not considered for further analysis. The high voltage sweep applied to the Ti electrode during the forming process generates the conductive filament inside the memristor. Supplementary Fig. 7.2 shows that the magnitude of the forming voltage is directly correlated to the initial resistance of the memristor. Consequently, devices with higher initial resistance, characterized by fewer defects, require a higher voltage to establish the conductive filament. Apparently, this trend holds true for all devices, irrespective of the device cross-section. To ensure that the measured initial resistance and subsequent resistive switching were not influenced by an unintentional TiOx layer at the interface (Ti layer susceptible to oxidation), devices fabricated without the hBN layer were measured (Supplementary Fig. 8). These devices exhibit ohmic conduction with low resistance (50Ω–180Ω), confirming the absence of any substantial TiOx layer.

After the forming operation, memristors exhibit bipolar resistive switching characteristics with positive voltage and negative voltage applied to the top Ti electrode for SET (transition to lower resistance) and RESET (transition to higher resistance) operations, respectively. Figure 2a shows the representative DC current–voltage (I–V) sweeps of the memristor device with a current compliance limit of 100 μA. Additional I–V sweeps measured from multiple devices are presented in Supplementary Fig. 9. Data collected from 200 such I–V traces were analyzed to quantify the variation in the device characteristics. Statistical analysis shows that the minimum separation between the high-resistance state (HRS) and low-resistance state (LRS), is ~50x (Fig. 2b), considering variations. Similarly, the variations in the SET and RESET voltages are characterized by Gaussian distributions with mean values (μ) of 1.73 V and −0.85 V, respectively (Fig. 2c). The SET voltage distribution displays higher dispersion (standard deviation (σ) of 0.5 V) mainly attributable to the device-to-device variations (see Supplementary Fig. 10 for discussion). In comparison, the RESET voltage distribution is rather tight with σ of 0.27 V. This variation in switching voltage can be further reduced by ensuring high crystalline quality of the synthesized 2D material and by employing superior device fabrication methods that can reduce electrode surface roughness.

To study the transport mechanisms responsible for the switching process, SET I–V sweep was plotted on a log-log scale

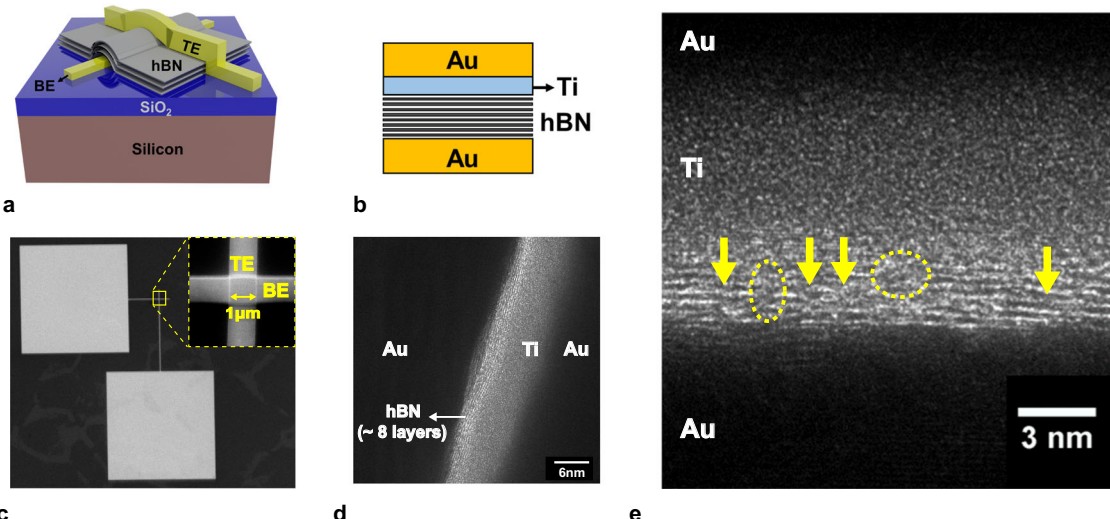

**Fig. 1 | Device structure of the fabricated 2D hBN memristor. a** Schematic drawing of the memristor device. **b** Cross-section schematic showing the metal contact used for top (titanium) and bottom electrodes (gold). **c** SEM image of the device after fabrication (inset) zoomed image where top and bottom electrodes cross. **d** Phase contrast TEM image of the device cross-section showing 8 layer thick hBN sandwiched between metal electrodes. **e** Magnified TEM cross-section showing the layered structure of hBN with single-atom defects (arrows) and few atom wide amorphous regions (dashed circles).

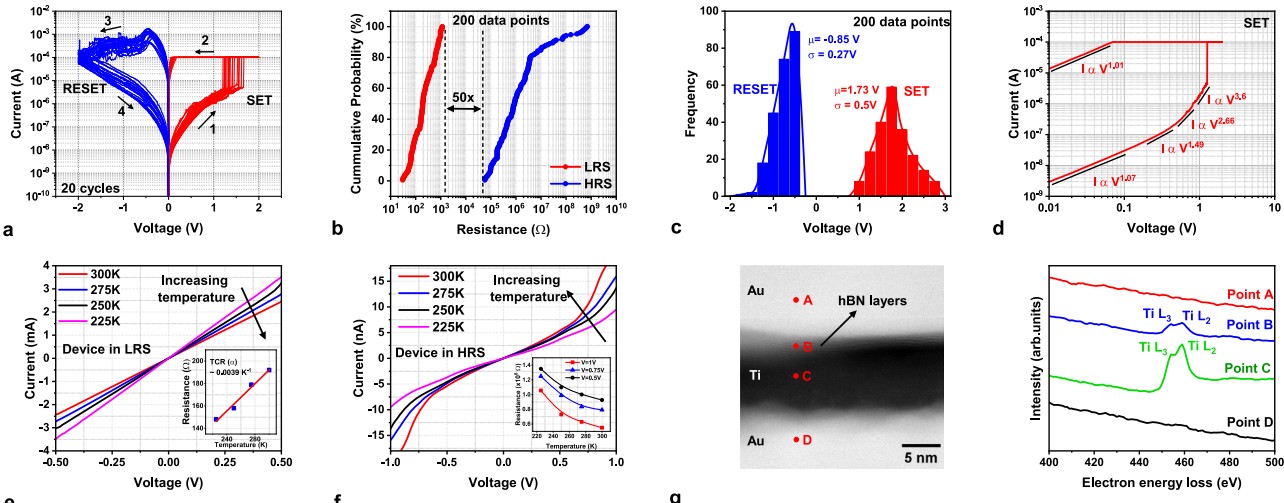

**Fig. 2 | DC characterization of the 2D hBN memristor. a** DC I−V sweeps of the hBN memristors shown for 20 consecutive cycles (SET curves shown in red traces and RESET curves shown in blue traces). **b** Cumulative probability distribution of the resistance in HRS and LRS states extracted from DC sweeps at $V_{READ}$ = 0.1 V collected from 200 I−V traces. **c** SET/RESET voltage distribution using 100uA compliance current. **d** SET I−V curve plotted on log-log scale shows regions with varying slope in HRS and linear slope in LRS. **e** I−V sweep of the device in LRS measured at various operating temperatures (225 K (pink curve), 250 K (black curve), 275 K (blue curve), 300 K (red curve)); (inset) resistance extracted at $V_{READ}$ = 0.5 V plotted vs.

temperature. The memristor in LRS has a positive temperature coefficient of resistance indicating the formation of metallic filaments (**f**) I−V sweeps of the device in HRS measured at various operating temperatures (225–300 K); (inset) resistance extracted at $V_{READ}$ = 0.5 V (black curve),0.75 V (blue curve),1 V (red curve) plotted vs. temperature. Reducing resistance with increasing temperature suggests conduction through insulating medium. **g** TEM cross-section illustrating the locations where electron energy loss spectroscopy (EELS) profile was collected. Points A (red curve), D (black curve) are located inside Au electrode, point B (blue curve) is located inside hBN layer, and point C (green curve) inside Ti electrode.

(Fig. 2d). Device transitioning to LRS during SET traverses through conductance regions with varying slopes, indicating that the dominant charge transport mechanism varies: ohmic conduction with slope ~1, space-charge limited conduction (SCLC) in regions with slope ~2 and trap-fill limited SCLC in regions with slope >2[56,57]. Bidirectional voltage sweeps confirm that the change in slope is reversible and, therefore, not related to the filament formation (Supplementary Fig. 11). Further voltage increase results in an abrupt current jump (slope >10) suggesting the formation of an ohmic conductive filament (slope ~1). The current compliance prevents further expansion of the filament and protects the device from thermal breakdown. To corroborate the filamentary conduction mechanism, conductive atomic force microscopy (CAFM) measurements were conducted on a stressed device. The CAFM maps reveal isolated regions exhibiting high current, clearly indicating the formation of conductive filaments across the hBN layer (details in Supplementary Fig. 12). Likewise, the RESET I−V sweep also exhibits ohmic conduction in LRS and SCLC in the HRS, as seen in Supplementary Fig. 11. To understand the fundamental nature of the filament, I−V sweeps were measured at different operating temperatures. The resistance of the device in LRS increases with temperature suggesting the metallic nature of the filament (Fig. 2e). On the other hand, HRS resistance decreases with temperature indicating energy-activated conduction (Fig. 2f). In addition, Electron Energy Loss Spectroscopy (EELS) was employed to investigate the elemental composition of the filament. The EELS profiles gathered from four distinct points (A, B, C, D) on the TEM cross-section are plotted in Fig. 2g. A prominent Ti peak was observed at point C (located inside the Ti layer) while a faint peak was observed at point B (located within hBN layers). The presence of Ti peak at location B suggests titanium ion migration into the hBN layer, which contributes to the filament formation. Similar observations of titanium ion-constituted filaments have been reported in previous studies[55,58]. Overall, the temperature measurements and the EELS analysis collectively reinforce the existence of metallic filaments (composed of Ti atoms) in the 2D hBN memristor.

## Pulse voltage stress operations

The switching dynamics of the memristors were studied by applying a sequence of ultra-short voltage pulses ($T_{PULSE}$ ~ 2.7 ns) with a rise/fall time of ~ 350 ps. In our test setup, the voltage pulses were delivered to the device through an impedance (Z = 50 Ω) matched network, as schematically illustrated in Fig. 3a (additional details provided in the Methods section). A 50 Ω termination resistor was soldered onto the probe-tip (Fig. 3a), which minimized reflections and ensured maximum power transfer to the device (Supplementary Fig. 13). The resistance state of the device was measured by applying a constant DC offset ($V_{READ}$ = 0.1 V) to the pulse. This DC component of the current signal was amplified and measured on the oscilloscope.

Figure 3b shows the oscilloscope waveforms of the applied voltage pulse and the measured current during a SET operation. The induced current rises slowly compared to the voltage pulse and saturates at ~700 ps which can be characterized as the SET switching time of the memristor. The I−V characteristics extracted from the transient waveforms in Fig. 3b clearly demonstrate the resistive switching behavior (Fig. 3c). To verify that the memory window observed in Fig. 3c originates from memristor resistive switching and not from system parasitics, we apply similar voltage pulses to discrete resistors (Supplementary Fig. 14). Further, the successful SET operation was also confirmed through the DC read signal (Supplementary Fig. 15). Figure 3d shows the transient waveforms of the voltage and current pulses for the RESET operation. Initially, since the device is in LRS, the current increases, reaches a maximum, and gradually decreases as the filament dissolution proceeds. This initial current surge should not be confused with parasitic displacement current since the 50Ω shunt resistor placed on the probe tip provides an effective path to dissipate parasitic charges. The successful resistive switching during the RESET operation is confirmed by the I−V plot (Fig. 3e) and the DC read signal (Supplementary Fig. 15).

## Statistical analysis of switching dynamics

The filament formation and dissolution are stochastic processes determined by several factors apart from the electric field, such as

energy generation/dissipation, defects in the 2D film, and the interface with the electrodes. Hence, statistical analysis of the switching dynamics was employed to gain insights into the resistive switching mechanism. This study was conducted by applying identical voltage pulses ($V_{PULSE}$ ~ 2.75 V, $T_{PULSE}$ ~ 2.7 ns) and analyzing changes in the switching behavior across different devices/cycles. Data from the measured current pulses display a wide range of intrinsic switching

times (600 ps to 2.5 ns—see Supplementary Fig. 16 for details) as seen from Fig. 4a (voltage pulses traces omitted for clarity). The distribution of switching times (extracted from each current trace) has a mean value of 1.32 ns and a standard deviation of 670 ps (Fig. 4b). Notably, ~46% of the devices exhibit sub-nanosecond switching.

Further, to understand the effect of filament morphology on switching dynamics, we studied the correlation between switching

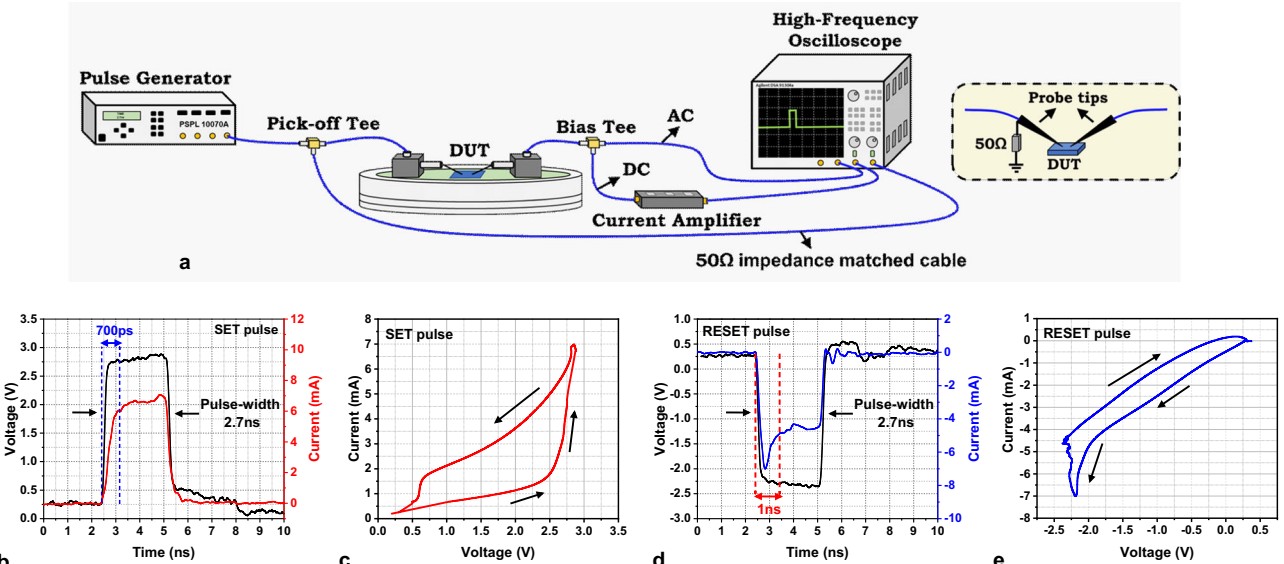

**Fig. 3 | Ultra-fast pulse characterization setup and waveforms. a** Schematic drawing of the high-frequency pulse test setup; (right) zoomed schematic of the probe tips with 50 Ω termination for maximum power transfer .**b** Applied voltage pulse (black trace) and measured current (red trace) waveform for SET operation. The applied pulse has a pulse-width of 2.7 ns and the device switches in about

700 ps, **c** *I–V* plot of the data in (**b**) shows change in resistance during the applied pulse. **d** Applied voltage pulse (black trace) and measured current (blue trace) waveform for RESET operation (**e**) *I–V* plot of the data in (**d**) shows change in resistance during the applied pulse.

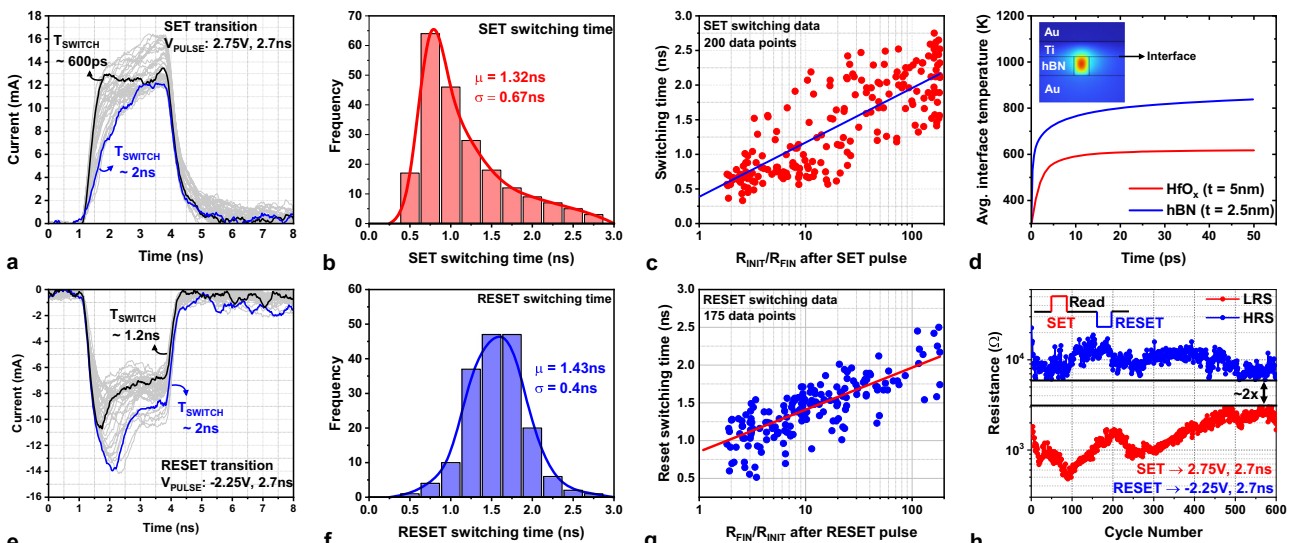

**Fig. 4 | Statistical analysis of transient switching characteristics using ultra-fast pulses. a** Measured current waveforms on 30 different devices under identical applied voltage pulses ($V_{PULSE}$ → 2.75 V, 2.7 ns) during SET operation. **b** SET switching time distribution created from 200 measurements with a mean value of 1.32 ns. **c** Correlation between SET switching time and resulting relative change in resistance values. Large resistance change requires a longer switching time. **d** Transient response of average interface temperature for 2.5 nm thick hBN and 5 nm thick HfOx, obtained through COMSOL simulation (inset) snapshot of

temperature heatmap. **e** Measured current waveforms on 30 devices under identical applied voltage pulses ($V_{PULSE}$ → −2.25 V, 2.7 ns) during RESET operation. **f** RESET switching time distribution created from 175 measurements with a mean value of 1.43 ns. **g** Correlation between RESET switching and resulting relative change in resistance after the RESET pulse. **h** Switching endurance measurement of single device demonstrates 600 cycles with 2.7 ns pulses (SET data points shown in red and RESET data points shown in blue).

time and resistance change induced by the SET pulse ($R_{INIT}/R_{FIN}$ where $R_{INIT}$ is the initial state resistance and $R_{FIN}$ is the final state resistance) (Fig. 4c). Here, the resistance values ($R_{INIT}$, $R_{FIN}$) were obtained by DC measurements before and after the SET pulse. For this study, devices were programmed (by leveraging the gradual RESET characteristics shown in Fig. 2a) such that the initial resistances were in the $10^4$–$10^5$ Ω range and the set pulse results in $R_{FIN} < 10^4$ Ω. The switching time varies among devices even with the same $R_{INIT}$ and $R_{FIN}$, highlighting the stochastic nature of filament growth. However, a clear upward trend indicates that larger resistance change (which entails more structural modifications) requires longer switching times.

The switching mechanism in 2D memristors is typically based on the formation and dissolution of conductive filaments[55,58]. Although a recent study[59] reports non-filamentary conduction in hBN memristors, our devices exhibit filamentary switching, as confirmed by the ohmic behavior in LRS and positive TCR in LRS. The filament growth in memristors proceeds through vertical propagation (around defects or weak spots), followed by lateral expansion[60–62]. The current flowing through the initially formed narrow filament (single atom or few atoms wide) raises the temperature inside the memristor through the Joule heating phenomenon. The generated temperature helps the Ti ions surmount the energy barrier (Arrhenius dependence) and migrate into the hBN layer, further propelling the filament growth[61–63]. To understand the fundamental factors responsible for the ultra-fast switching in 2D memristors, we conducted transient thermal analysis using a finite element method-based physics solver (COMSOL). The ultra-thin 2D hBN layer increases the electric field as well as the temperature in the memristor, both of which promote rapid filament growth. Moreover, the high in-plane thermal conductivity of the hBN layer[64,65] (~ 35x higher than transition metal oxides[66–68]) rapidly spreads the heat generated in the filament (joule heating) to the Ti/hBN interface (detailed analysis presented in Supplementary Fig. 17). The high

temperature at the Ti/hBN interface facilitates Ti ion release from the electrode into the hBN switching layer. Figure 4d compares the average interface temperature for our devices and HfO$_x$ memristors with 5 nm thickness (typical thickness for oxide memristors). Evidently, the interface temperature rises rapidly in our devices compared to thicker oxide memristors, consequently resulting in faster resistive switching.

Similar to SET operation, RESET also displays a range of intrinsic switching times (Fig. 4e). The RESET operation is slower than the SET operation (distribution μ = 1.43 ns, σ = 400 ps) with only ~8% of the devices exhibiting sub-ns switching (Fig. 4f). As in SET, a larger resistance increase is associated with a longer switching time (Fig. 4g). Here, the devices have $R_{INIT} \sim 10^3$–$10^4$ Ω, and the RESET pulse results in $R_{FIN} > 10^4$ Ω. Further, the switching endurance test demonstrates one of the highest reported values (600 cycles) with the shortest applied pulses (~2.7 ns) in 2D memristors (Fig. 4h). Consistent switching between resistance states is very promising since it demonstrates the potential of 2D memristors for ultra-fast switching memory applications.

## Role of Joule heating on filament stability

Several studies have reported that Joule heating promotes filament dissolution in TMO memristors[69–71] as well as in 2D memristors[72,73]. Especially in devices with metallic filaments, large LRS current generates considerable heat (600–1200K) that accelerates the out-diffusion of metal ions[74,75]. The joule heating can be estimated by the energy generated inside the memristor, which can be calculated by integrating the product of transient voltage and current pulses (obtained from Fig. 4a, e) over time ($E = \int_0^t V(t)I(t)dt$) (see Supplementary Fig. 18 for details). This energy can be roughly segregated into switching energy and excess energy, as shown by the schematic in Fig. 5a. Switching energy contributes to the resistive switching process while excess energy continues to dissipate throughout the device after the switching process is complete. The switching energy driving the

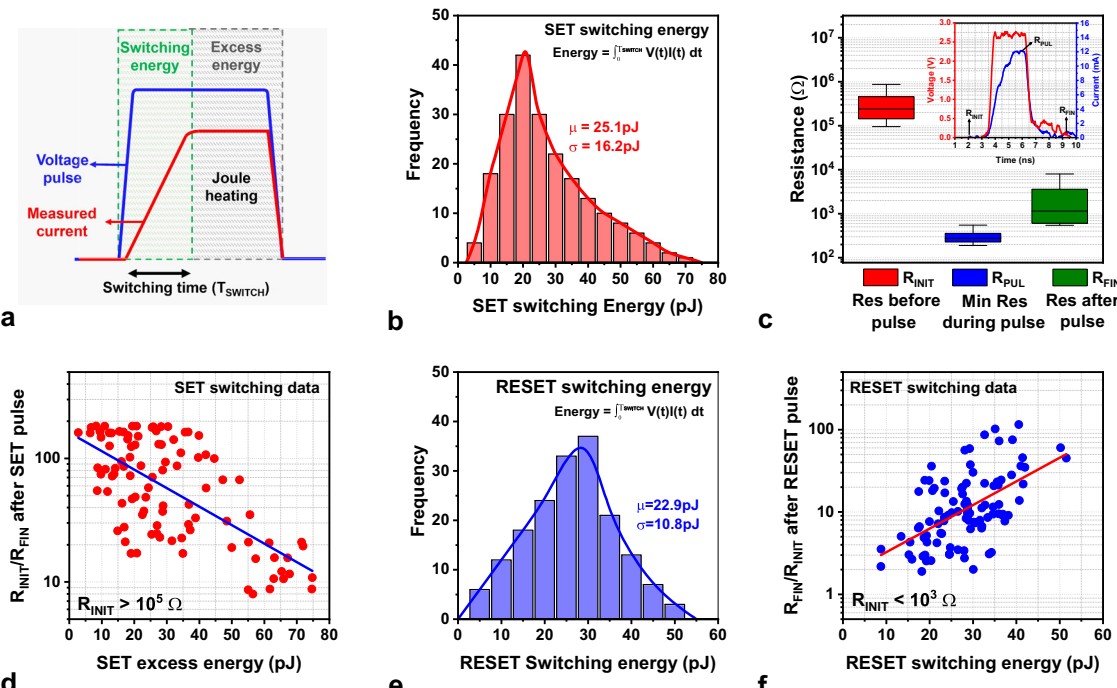

**Fig. 5 | Statistical analysis of memristor switching energy. a** Schematic representation of the partition between switching energy and excess energy. **b** SET switching energy distribution created from 200 measurements with a mean of 25.1 pJ. **c** Resistance distribution of $R_{INIT}$ (red box plot), $R_{PUL}$ (blue box plot), and $R_{FIN}$ (green boxplot) (inset) waveform illustrating the time instance at which the resistances were measured. The distribution clearly indicates the filament

resistance increases after pulse termination. **d** Relative resistance change after SET pulse variation vs excess energy expended in the device. Since excess energy is proportional to the Joule heating in the device, higher excess energy results in narrowing of the filament. **e** RESET switching energy distribution created from 175 measurements with a mean of 22.9 pJ. **f** Relative resistance change after RESET pulse vs switching energy.

SET operation has a mean of 25.1 pJ and a standard deviation of 16.2 pJ (Fig. 5b). On the other hand, the mean excess energy (μ - 39.3 pJ) is higher than the switching energy attributable to the large LRS current flowing through the device.

For the SET transition, the memristor resistance distribution measured after the pulse was observed to be higher than the resistance during the pulse (Fig. 5c). Ideally, the resistance should have remained unchanged as the memristor in LRS exhibits linear ohmic conduction (based on Fig. 2d). Identical pulses applied to discrete resistors displayed no change in resistance during and after the pulse (Supplementary Fig. 19), confirming that the test setup parasitics do not contribute to the resistance change. We hypothesize that the joule heating-assisted filament narrowing is responsible for the observed resistance increase. To understand how this phenomenon affects the final resistance state ($R_{FIN}$), the correlation between excess energy dissipated in the device (proportional to filament temperature) and the resistance change ($R_{INIT}/R_{FIN}$) was studied (Fig. 5d). For this study, devices with similar initial resistance ($R_{INIT} > 10^5\ \Omega$) were chosen to eliminate the impact of initial filament morphology. Evidently, higher excess energy results in lower resistance change, indicating significant filament narrowing caused by enhanced joule heating. Supplementary Fig. 20 presents a further analysis of resistance change induced by Joule heating.

The phenomenological model based on the Arrhenius relationship is a widely accepted model which satisfactorily captures the filament growth dynamics in TMO memristors[59–62]. Therefore, in this study, we employ a modified version of this model to capture the growth dynamics as well as the Joule heating-induced filament narrowing in our 2D memristors. In the classic model, time dependency of the filament growth is described by the Arrhenius relationship $\left(\frac{d\Phi}{dt} = A_1 \exp\left(-\frac{Ea0 - \alpha qV}{kT}\right)\right)$[59–62], where $\Phi$ is the filament diameter and $E_a$ is the activation energy and $\alpha qV$ accounts for the barrier lowering due to the applied voltage. This simple relationship cannot capture the filament narrowing observed in our devices. Therefore, an additional term $\left(\frac{d\Phi}{dt} = -A_2 \exp\left(-\frac{Ea}{kT}\right)\right)$ was included that accounts for Joule heating-induced filament dissolution (see Supplementary Fig. 21 for detailed analysis). Based on this model, the following observations can be deduced: During the switching energy phase, when the Joule heating is insignificant (low current in the memristor), the filament growth rate dominates, resulting in filament expansion. On the other hand, during the excess energy phase when Joule heating is significant (high current in the memristor), the filament growth rate balances the filament dissolution rate resulting in current saturation. Finally, during the voltage pulse falling transition, the filament growth rate (determined by $V$, $T_{CF}$) decreases faster than filament dissolution rate (determined only by $T_{CF}$), thereby resulting in narrowing of the filament. A self-consistent electro-thermal solver using the proposed model clearly captures the filament narrowing phenomenon (see Supplementary Fig. 21) which closely matches with the experimental data presented in Fig. 5c. In addition to the proposed electro-thermal model, the resistance evolution of the memristor can be adequately captured through a simple empirical model. The Arrhenius relationship with an added filament enlargement factor $\Phi^n$ $\left(\frac{d\Phi}{dt} = A \frac{\exp\left(\frac{-Ea0}{kT}\right)}{\Phi^n}\right)$ produces a good fit to measured data (Supplementary Fig. 22). In this model, the temperature is considered constant, which significantly reduces the computational cost. Such models are particularly beneficial for implementing large circuits using the SPICE simulator.

Mean RESET switching energy (μ - 22.9 pJ) is slightly lower than the SET switching energy, as seen from Fig. 5e. Unlike SET operation, Joule heating in the RESET operation is dominant during the switching energy phase (wider filaments supporting higher currents). The resistance change ($R_{FIN}/R_{INIT}$) after the RESET pulse clearly increases with increasing switching energy, as seen in Fig. 5f. A similar trend of increasing resistance change can be observed from the total energy plots presented in Supplementary Fig. 23. Further, to confirm the role of heat on filament stability, we conducted retention studies at elevated temperature (350–700K). The experiments show that memristors in LRS or HRS display resistance increase with time, validating the thermally-driven filament dissolution (Supplementary Fig. 24). Further, the LRS retention time shows an exponential dependence on the temperature, characterized by an activation energy of 0.668 eV. Extrapolating the graph shows that the memristors in LRS would exhibit a retention time of 10 years at room temperature (Supplementary Fig. 24).

## Ultra-fast (120 ps) switching memristors

The memristive devices fabricated using thick hBN (7–8 layers) exhibited varying ranges of intrinsic switching times, with the tail-bits reaching as low as 600 ps. To evaluate the lowest possible switching time, we tested the devices using ultra-short pulses ($T_{PULSE}$ ~ 120 ps). These ultra-short pulses were generated through a step-recovery diode-based comb generators. Although thick hBN devices exhibit resistive switching under ultra-short pulses, the switching was inconsistent. The devices generally fail to switch in multiple cycles, especially in the RESET operations (Supplementary Fig. 25). To understand the impact of hBN layer thickness on switching, devices with thin hBN films (3-4 layers) were fabricated. These devices have lower initial resistance due to reduced thickness and do not require initial forming operation. Supplementary Fig. 26 shows repeatable switching using DC voltage sweeps in these devices. These devices exhibited resistive switching with ultra-short voltage pulses of 120 ps FWHM (full width at half maximum) (Fig. 6a, b). This is the fastest ever reported switching time observed in 2D memristors surpassing the previous best report of 700 ps in MoS$_2$ memristors[47]. Note that higher amplitude voltage pulses (~4 V) were required for successful SET and RESET operation compared to ~2.75 V of longer nanosecond pulses (Fig. 3b, d). The devices consumed <2pJ for SET/RESET transitions (see Supplementary Fig. 27 for details) that is among the lowest reported switching energies in 2D material-based memristors. Repeated cycling using 120 ps pulses shows consistent switching (100 cycles) with a memory window of ~1.7x (Fig. 6c). This is the highest reported endurance using sub-ns pulses, exceeding the best-reported value of 20 cycles observed in the TMO memristor[38] (see Fig. 6d and Supplementary Table. 1 for details). Supplementary Table 2 compares the endurance performance of our devices with other 2D memristors.

## Discussion

In summary, we fabricated Ti/hBN/Au memristors that exhibit ultra-fast switching characteristics when characterized using short voltage pulses ($T_{PULSE}$ < 3 ns). Unlike other ultra-fast memristors, our devices with thicker hBN (7–8 layers) exhibit high cycling endurance (~600 cycles) using short 2.7 ns voltage pulses with a mean switching time of 1.43 ns. On the other hand, thin hBN (3–4 layers) devices switch under ultra-short pulses of 120 ps, which is the fastest reported switching speed among 2D material-based memristors. Finite-element-analysis-based thermal simulations provided key insights which help explain the fundamental reason responsible for the observed ultra-fast switching in 2D memristors.

Statistical data collected from several devices revealed useful correlations between switching time-resistance ratio and switching energy-resistance ratio, providing a practical knob to tune device characteristics. Furthermore, the unique experimental methodology employed in this work allowed us to study the dynamic evolution of resistance (during the pulse) as well as the change in resistance (before and after the pulse). To the best of our knowledge, this is the first instance where such an approach has been adopted. Correlating these data points revealed a significant presence of joule heating-induced filament narrowing in our devices. The impact of Joule heating on filament stability was studied using a combination of statistical analysis, retention testing, and electrothermal simulations. Moreover, an electro-thermal model was developed which accurately captures the

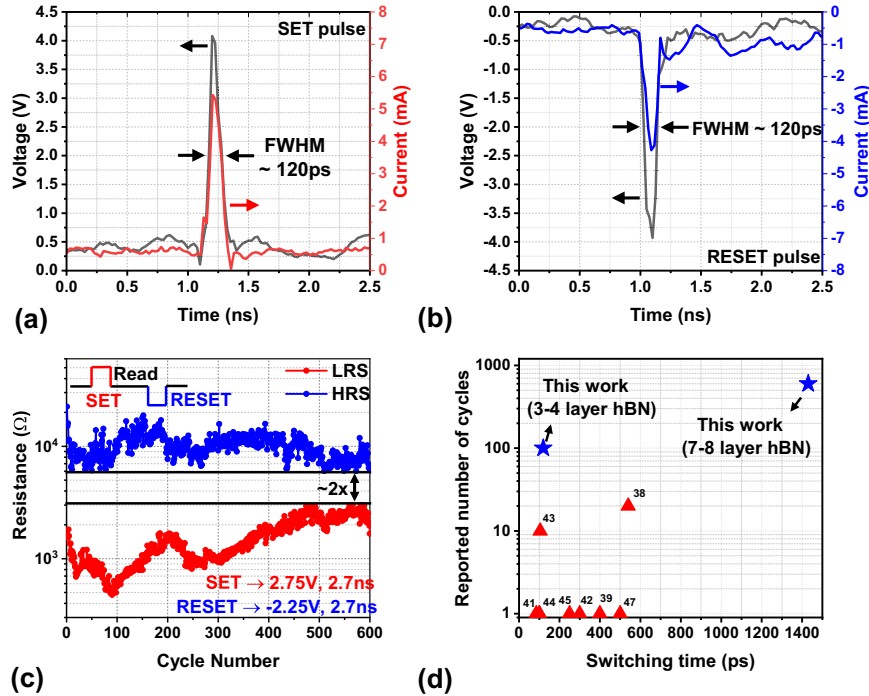

**Fig. 6 | Demonstration of highest switching speed in 2D memristors. a** Applied SET voltage pulse (gray trace) and measured current (red trace) for thin hBN (3–4 layers) memristor. The pulse-width of the voltage pulse is about 120 ps. **b** Applied RESET voltage pulse (gray trace) and measured current (blue trace) using 120 ps pulse. **c** Endurance cycling between LRS and HRS of the memristor with ultra-short pulses. The device shows 100 consecutive cycles of consistent switching, which is the highest ever reported endurance using ultra-short pulses. **d** Plot benchmarking the performance of the devices presented in this article with other TMO and 2D material memristors.

transient switching characteristics including secondary effects such as Joule heating. Overall, this study unveils the true potential of 2D material memristors for future high-speed computing, storage, and RF applications.

## Methods

### Hexagonal boron nitride synthesis

Few layer hBN stacks were grown by Chemical Vapor deposition (CVD) on commercially available copper foils (25um thick, 99.8% purity, annealed, uncoated – Alfa Aesar Corporation). The foils were electro-polished using phosphoric acid-based solution to reduce the surface roughness and eliminate the striations typical of commercially produced copper foils. The copper foil was folded into an enclosure and placed in the center of the quartz tube. The furnace temperature was ramped-up and the substrates were annealed at 1030 C for 25 min in $H_2$ environment (30sccm flow rate). After annealing the substrates, the ammonia-borane precursor (placed in a separate heating zone – see Supplementary Fig. 1) was heated to 92C-95C in a water bath. At this temperature, ammonia-borane decomposes to borazine, monomeric aminoborane and hydrogen that are introduced into the chamber through $H_2$/Ar carrier gas. The chemical species adsorbed onto the copper surface synthesize the desired few-layer hBN stacks. During the growth, $H_2$/Ar flow rate was 80sccm/14sccm and the overall chamber pressure reached ~700mT. The duration of the growth varied ranging from 10 to 12 min Following the growth, the chamber was rapidly cooled (~30 min) through a natural cooling process and the gas flow was maintained throughout the cooling process. The layered structure of the grown hBN films was confirmed using TEM imaging (Fig. 1 and Supplementary Figs. 2, 3).

### hBN film transfer

The grown hBN films were transferred from copper foil onto Si/SiO$_2$ (285 nm) substrate for device fabrication. For enabling the transfer process, the copper foils with hBN films were spin-coated with PMMA (polymethyl methacrylate) at 3000 rpm for 1 min followed by baking at 180 C for 1 min. The PMMA layer provides support and protection to the hBN films during the transfer process. The hBN films were transferred using $H_2$ bubbling transfer method using NaOH solution as the electrolyte. The spin-coated copper foil was used as the cathode and nickel substrate was used as the anode for the electrolysis process. Applying 1.8 V to the anode generates $H_2$ gas bubbles at the cathode which leads to the delamination of hBN from the copper foil. The separated hBN/PMMA stack was rinsed several times to remove residues introduced during the transfer process and scooped up onto the target substrate with patterned bottom electrode. The sample with hBN/PMMA stack was stored in a vacuum chamber overnight to remove any water bubbles trapped between the hBN film and the substrate. Then the PMMA was stripped by leaving the sample in acetone for 2 hours. To further remove any PMMA residues, the sample was annealed at 200 C for 4 hours in a high vacuum chamber (1e-6 Torr). Successful transfer of the hBN film onto target substrate was confirmed by Raman and XPS spectroscopy, SEM and TEM imaging (Supplementary Figs. 2,3,4).

### Memristor device fabrication

The memristor device fabrication involves patterning the Bottom Electrode (BE) using e-beam lithography on Si/SiO$_2$ (285 nm) substrates, followed by e-beam evaporation of metal contacts (Au(70 nm)). The grown hBN film was transferred onto the BE using PMMA-assisted bubbling transfer method. The device fabrication was completed by patterning and depositing Top electrode (TE) metal (Ti(5 nm)/Au(70 nm)) such that titanium metal is directly in contact with hBN. The fabricated devices have varying cross-sectional area between 500 x 500nm$^2$ and 2 x 2 um$^2$. The final device structure was confirmed using optical and SEM imaging (Supplementary Fig. 6).

## Electrical DC characterization

DC characterization (Fig. 2a-d) of the fabricated memristors was conducted using a probe station (Cascade Microtech) and a semiconductor parameter analyzer (Agilent 4156 C). In all the measurements, the bias was applied to the TE with the BE grounded and the current compliance was typically set to 100uA. The temperature measurements (Fig. 2e, f) were conducted in a cryogenic probe station (Lakeshore FWP6) connected to an Agilent B1500 semiconductor parameter analyzer.

## Electrical pulse characterization

Pulse characterization (Figs. 3, 4, 5, 6) of the memristors was conducted in a custom in-house RF measurement setup. The nanosecond pulses ($t \sim 2.7$ ns used in Figs. 3, 4) were generated using a Picosecond Pulse Labs 10070A pulse generator. These ultra-short pulses were delivered to the memristors via an impedance (50 Ω) matched network to eliminate the reflections (see Supplementary Fig. 12). A 50Ω termination resistor was soldered onto the probe-tip to ensure maximum power transfer to the device as well as provide a quick path for dissipating parasitic charges. The voltage pulse delivered to the device was monitored through a pick-off tee that samples the signal with a 20 dB attenuation. The voltage pulse induces a current pulse that traverses downstream to the oscilloscope. Here the signal was split into AC and DC components using a bias-tee. The AC component of the signal was captured using a high-performance oscilloscope (Agilent DSA91304A−13 GHz, 40GSa/s). Since the input impedance of the oscilloscope is 50Ω, only current signals with amplitude > 200uA can be reliably captured. However, the memristors typically conduct 1uA-100uA current under read voltages ($V_{READ} \sim 100$mV-500mV). Therefore, to study memristor switching, the DC component of the signal was amplified using a current-to-voltage converter (Femto DHCPA-100) with a maximum operational frequency of 200 MHz. The switching in memristor was monitored by sampling the DC data before (500 ns) and after (4us) the pulse. The splitting of the signal allows us to study the dynamics of the memristor during switching (Figs. 3, 4a, b, e, f, 5b, e) as well as after the switching was complete (Figs. 4c, g, h, 5d, f, 6c). The ultra-fast pulse ($T_{PULSE} \sim 120$ ps) was generated by sending nano-second pulses through a step-recovery diode-based comb generator.

## Physical characterization

The quality of the synthesized hBN films was confirmed by Raman spectroscopy (532 nm laser−Renishaw InVia Raman spectrometer) and X-ray Photo-electron spectroscopy (XPS). The surface topography of the transferred hBN layers was characterized by Atomic Force Microscopy (Bruker Icon AFM). The final device structure was imaged using Optical Microscopy, SEM (Zeiss Neon 40 SEM) and TEM (Jeol neoARM Scanning TEM). The sample cross-section for TEM was prepared using focused ion beam (Thermo Scientific Scios 2 HiVac).

## Data availability

The authors declare that the data that support the findings of this study are available within the article and its Supplementary Information files. All other relevant data are available from the corresponding author upon request.

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

## Acknowledgements

This work was performed in part at the University of Texas Microelectronics Research Center, a member of the National Nanotechnology Coordinated Infrastructure (NNCI), which is supported by the National Science Foundation (grant ECCS-2025227). A.R. acknowledges support from the Science and Engineering Research Board, Govt. of India, Grant #SRG/2022/000788. It was supported in part by the Semiconductor Research Corporation grant LMD 3009.001.

## Author contributions

S.S.T.N., A.R. and S.K.B. conceived this work and prepared the manuscript. S.S.T.N. synthesized the 2D material, fabricated the memristor and performed electrical characterization. A.R. conducted the X-ray photoelectron spectroscopy measurement. S.S.T.N., D.V. and G.B. set up the ultra-fast pulse measurement setup and helped with analyzing the measurement data. M.C., K.C.M. and J.W. performed the Transmission Electron Microscopy imaging of the fabricated devices. S.S.T.N. and M.D. conducted the temperature based resistance measurement of the fabricated devices. S.S.T.N., A.R., A., J.V.S and J.P.K. were involved in analyzing the measurement results through electro-thermal model, phenomenological model and thermal simulations. I.R.G.D. performed the conductive AFM experiment and analyzed the result. All authors discussed and contributed to the analysis presented in the manuscript. S.K.B. supervised the study.

## Competing interests

The authors declare no competing interests.
