## [Peer Review File · Nature Communications]

REVIEWER COMMENTS

Reviewer #1 (Remarks to the Author):

This is a very good manuscript in the field of 2D-materials-based electronic devices for multiple reasons that are very clear to see. First of all, the authors fabricate small devices of $1\mu\text{m}^2$, and they present data from many devices (>160). Moreover, they do cross-sectional TEM to prove the structure of the device, and they do advanced electrical characterization (i.e., very few articles measure so short switching times). The conclusions are interesting and, more important, solidly proved.

However, there are several points that could be improved:

1 – Figure 2a gives bad impression. I would replace it by something like Figure S7a and then mention in the text that 8% of the devices were discarded because showed low resistance.

2 – Figure 2b is nice, but I'd like to see that for many devices. If not 160, then select the best 40-50 ones and put them together in the supporting information, indicating the number of cycles measured for each one. Similar to supplementary figure 20 of this paper: <https://www.nature.com/articles/s41928-020-00473-w>

3 – Figure 3c is a bit misleading, should give the ratio from the most-right red symbol to the most-left blue symbol, like in Figure 2d of this paper: <https://www.nature.com/articles/s41928-020-00473-w>

4 – I would like to see in the article the characteristics (fresh I-V curves) of 5-10 devices without the h-BN, to prove that what they are showing is not produced by the Ti film (which sometimes can easily form TiO_2 and show RS even without an insulating layer).

5 – Regarding the switching mechanism, it is fine for me and properly demonstrated. I would say that the authors could include the EELS data in Figure 2 to make it more powerful. Also, the authors should comment on the recent article demonstrating that non-filamentary RS in h-BN is possible (see <https://www.nature.com/articles/s41586-023-05973-1>)

6 – In the pulsed voltage stresses they should show at least one entire read/write/read/erase/read cycle, so that it can be confirmed that it switches well with cycles. Something like Extended Data Figure 8 in the paper <https://www.nature.com/articles/s41586-023-05973-1>

7 – The authors should discuss why the endurance is so low. They may show SEM images after the stresses, to see what happens. I guess the wires will melt because the currents are high. Please comment. In fact, I think they could reach higher endurance changing the design of the masks. There is no need to make such long narrow lines, one can reduce the width progressively from the pad as one approaches to the Crosspoint section. This reduces current density and slows down electromigration.

8 – I wonder why the authors don't make TEM image with higher resolution to see more clearly the structure of the devices. It would be good if they can provide something like in this paper: <https://www.nature.com/articles/s41699-022-00341-5>

I think that, after these minor revisions, the article should be published because it advances the state-of-the-art in this field.

Reviewer #2 (Remarks to the Author):

This paper presents a study on hBN-based memristors that exhibit an ultra-fast response of 120 ps. To investigate the memristor switching mechanism, the authors employed statistical analysis. The work includes a comprehensive analysis of the hBN memristor model, covering filament growth, joule heating effect, and the correlation between switching energy and switching ratio. In addition, they conducted endurance measurements with ultra-fast pulses and employed statistical analysis to demonstrate the device's potential for high-frequency applications. Despite the potential significance of the work, the novelty and contribution are inadequately described and do not meet the publication standards of Nature Communications. The detailed reasons are listed below:

Major issues:

1. In the introduction, the authors noted that there has been relatively little attention devoted to studying the switching dynamics of memristors. However, numerous studies have explored the dynamic mechanisms in both metal-oxide-based and hBN-based memristors. For instance, TEM has been used to observe the electrochemical dynamics of nanoscale metal conductive filaments in metal-oxide-based memristors, and a model has been established to consider the applied electric field, metal ion diffusion, and filament shape based on dynamic metal cluster nucleation and growth. [1,2] In the case of 2D hBN-based memristors, the dynamic analysis of nano-conductive filaments has also been investigated,

including the effect of temperature. [3,4] Additionally, a Monte Carlo simulation has been shown to simulate filament growth and erase dynamics. [5]

2. In Figure 2a, have the initial currents measured from devices with different areas been plotted together? If not, the authors should clearly state the device area that used in this figure. If initial currents have been plotted together, the authors should include a legend specifying the device area, as this parameter is critical in determining the high-resistance state (HRS) initial current. A similar issue is observed in Supplementary Figure 7a, where the authors plot the correlation between the forming voltage and the initial current. In this case, the authors should keep the device area constant.

3. In Fig. 2d, the device cycle-to-cycle variation is much higher than other hBN-based memristors.[6] The range of set voltages is wide, spanning from 1 to 3 V, creating significant difficulties in selecting memristor devices for cross bar arrays. Furthermore, device-to-device variation exacerbates this issue. To understand the cause of this considerable variation, the authors should conduct further analysis.

4. In the study on dynamic filament formation (see Fig. 4 and Supplementary Fig. 10), the authors employed a conventional model that takes into account the effects of temperature, the heat equation, and the Arrhenius relationship for filament growth. It should be noted, however, that this model was originally developed for oxide memristors. [7,8] As such, it is unclear how the model in this work differs from the one used for metal oxide, and how it specifically contributes to understanding the filament growth dynamics in hBN.

5. The authors need to provide a clear explanation of the fundamental reason for the ultra-fast switching behavior observed in their memristor, as compared to other fast-switching memristors discussed in previous studies.[9,10] To achieve this, the authors should provide a detailed explanation in terms of their model and material properties.

6. It would be beneficial if the authors could compare the fitted filament diameter obtained from their model with experimental results obtained from techniques such as C-AFM or HRTEM, which can directly measure the filament diameter.

7. The authors should discuss more metrics or guidelines for measuring the pulse characteristics of their memristor. In Figures 4 and 5, the authors have presented statistical analysis of the transient pulse response and established correlations between switching ratio, switching time, and switching energy. It shows that the switching energy and switching time is highly related to the switching ratio before/after pulse. To better understand the performance of their ultra-fast switching device, the authors should discuss key performance metrics for fast-switching devices and the differences between normal

endurance (with pulse widths of tens of microseconds) and ultra-fast pulse endurance (with sub-nanosecond pulses).

8. The authors demonstrate 600-cycle endurance with ultra-fast pulses, which is higher than other fast memristors' endurance results. However, it is important to note that other memristors can achieve endurance up to 108 cycles with pulse widths of microseconds. Therefore, to prove the stability of their device, the authors should provide endurance results with a higher cycle number.

Minor issues:

1. There are some typos in the manuscript. (Line 81, the SET should be transition to lower resistance instead of lower conductance)

2. The image quality is poor in the manuscript, some of the images cannot be viewed clearly. (Figure 2g, 2h)

3. Scale bar labeling is confusing. In Supplementary Fig. 3a, the sample electrode width seems to be larger than 2 μm . However, the maximum electrode width is 2 μm in this work.

References list:

[1] Y. Yang, P. Gao, L. Li, X. Pan, S. Tappertzhofen, S. Choi, R. Waser, I. Valov, W. D. Lu, *Nat. Commun.* 2014, 5, 1.

[2] Y. Yang, P. Gao, S. Gaba, T. Chang, X. Pan, W. Lu, *Nat. Commun.* 2012, 3, 1.

[3] M. Lanza, F. Palumbo, Y. Shi, F. Aguirre, S. Boyeras, B. Yuan, E. Yalon, E. Moreno, T. Wu, J. B. Roldan, *Adv. Electron. Mater.* 2022, 8, 2100580.

[4] K. Zhu, X. Liang, B. Yuan, M. A. Villena, C. Wen, T. Wang, S. Chen, F. Hui, Y. Shi, M. Lanza, *ACS Appl. Mater. Interfaces* 2019, 11, 37999.

[5] F. Qian, R.-S. Chen, R. Wang, J. Wang, P. Xie, J.-Y. Mao, Z. Lv, S. Ye, J.-Q. Yang, Z. Wang, Y. Zhou, S.-T. Han, *IEEE Trans. Electron Devices* 2022, 69, 6049.

[6] S. Chen, M. R. Mahmoodi, Y. Shi, C. Mahata, B. Yuan, X. Liang, C. Wen, F. Hui, D. Akinwande, D. B. Strukov, M. Lanza, *Nat. Electron.* 2020, 3, 638.

[7] P. Huang, X. Y. Liu, B. Chen, H. T. Li, Y. J. Wang, Y. X. Deng, K. L. Wei, L. Zeng, B. Gao, G. Du, X. Zhang, J. F. Kang, *IEEE Trans. Electron Devices* 2013, 60, 4090.

[8] D. Ielmini, IEEE Trans. Electron Devices 2011, 58, 4309.

[9] B. J. Choi, A. C. Torrezan, K. J. Norris, F. Miao, J. P. Strachan, M.-X. Zhang, D. A. A. Ohlberg, N. P. Kobayashi, J. J. Yang, R. S. Williams, Nano Lett. 2013, 13, 3213.

[10] B. J. Choi, A. C. Torrezan, J. P. Strachan, P. G. Kotula, A. J. Lohn, M. J. Marinella, Z. Li, R. S. Williams, J. J. Yang, Adv. Funct. Mater. 2016, 26, 5290.

Reviewer #3 (Remarks to the Author):

This manuscript demonstrates the first sub-nanosecond switching of a 2D hBN-based memristor. In addition to novel switching data, a compelling thermal model is used to understand the nature of the filament during switching in this material. The statistical analysis of switching over many cycles in 30 devices is uniquely useful. Overall, the methods used in this work are sound and I was particularly happy to see the careful techniques like the control resistor experiment in S-Fig 11. It is well written and on a topic of interest to the community. Previous works are well referenced. I recommend publication after considering a few potential points for improvement:

--It would be useful if the authors could elaborate on the proposed switching mechanism. The implication that a Ti filament is responsible for switching (lines 99-103) a very interesting point, but only briefly mentioned. If cationic motion of Ti is responsible for the filament, perhaps this is a differentiating feature of 2D-based memristors from the traditional TMO memristors which are (typically) thought to be modulated by oxygen vacancy anions. I wonder if the physical insight here is applicable to traditional TMO memristors.

--Supplementary Table 1 is a useful reference comparing switching in standard TMO and 2D memristor. I am wondering if, based on this work, the authors can further comment on fundamental differences between switching in traditional versus 2D materials. I realize there is quite a bit of interest in the novelty of 2D materials, but am wondering if there are fundamental advantages of the 2D stack. My impression from this work and refs in the Table is that that 2D memristors have similar switching speed and energy, but the endurance may be somewhat worse.

--The resistance evolution in Fig 4d does not appear to align with the level described in the text. On lines 154, the initial range is 10k-100kohms, but in Fig 4d, we see the evolution of resistance in a much lower

range, from about 700-1000 to 200 ohms. If the authors have data showing SET switching starting from the higher range, that would be of interest.

--Minor: In general the variables are carefully defined for the model in the supplement, but I would recommend adding definitions for supplementary Eqn 6.

--Minor: The plot text on several figures is too small and hard to read (esp insets).

Reviewer #4 (Remarks to the Author):

The manuscript by Nibhanupudi et. al. represents the memristor device prepared by thin hBN layer and Ti and Au electrodes. Experiments present successful fabrication of device and memristance characterization as well as pulse characteristic including statistical analyses. Theory presents formation of filaments in a thermo-static medium. In my opinion this work is a valuable study in memristor but can not advice it for publication in Nature Communication. There are major list of comments in following

Theory: In my opinion the theoretical model is a valuable work but cannot be accepted as a close model of what happens in experiments. For example, 3 nm filament has very similar thickness to hBN layer. The materials of filament is not clear. There might be a population of filaments at the interface to electrode with different sizes. There is a complicated heat bath (generation and dissipation) that is not argued. What are authors looking for and what consequences are expected with such a model is not clear. Way VCM and TCM are not modeled and only attention is made on filament formation.

Experiment: There is no motivation to accept the reported record of fast switching in this layer as there is not a big change w.r.t. the previous studies. The finding of paper is not clear. What is new? statics? fast switching? thermal and heat model?

Technical points: intro: There is repeated theme of fast electronics or switching application in different paragraphs and there is no focus. There is not a coherent introduction that what are authors plan to present in the paper. experiment: i) a high resolution image of filament formation (for different biases) is lacking, ii) in the calculation of power, since current and thus impedance changes, how authors are sure to have a delivery of power to/from different resistance loads? Could part of power be reflected? What are high risk points that might not be considered?

Ultra-fast switching memristors based on two-dimensional materials

Author response to reviewer comments

Firstly, we would like to thank all the reviewers for taking the time to review and provide valuable feedback to improve the quality of our manuscript. We have updated and modified the manuscript to reflect the suggestions. Please see below the detailed point-by-point response for each comment.

Reviewer1

Figure 2a gives bad impression. I would replace it by something like Figure S7a and then mention in the text that 8% of the devices were discarded because they showed low resistance.

Thank you for the comment. As suggested, this figure has been moved to Supplementary Information (Fig.7) and we state the device yield (92%) in the main text.

Figure 2b is nice, but I'd like to see that for many devices. If not 160, select the best 40-50 ones and put them together in the supporting information, indicating the number of cycles measured for each. Similar to supplementary figure 20 of this paper: <https://www.nature.com/articles/s41928-020-00473-w>

Figure R1.1 DC IV characteristics collected from 12 different devices.

As suggested, we included measured DC electrical characterization data with a current compliance limit of 100uA. Although 160 devices were tested for initial resistance measurements, only 30 were tested for DC cycling measurements. The rest were allocated for AC testing, temperature measurements, retention testing, and TEM characterization. Here, we include data collected from 12 devices (Fig.R1.1). This figure has been included as Fig.9 in the supplementary information.

Figure 3c is a bit misleading, it should give the ratio from the most-right red symbol to the most-left blue symbol, like in Figure 2d of this paper: <https://www.nature.com/articles/s41928-020-00473-w>

Thank you for the comment. In the original manuscript, we included the ratio between LRS and HRS at the 50% percentile point, as reported in some earlier studies [1-2]. However, we agree with

your comment that this method of reporting the resistance window does not capture the worst-case scenario. Therefore, as suggested, we have updated the figure (Fig.2b main text) with the ratio between the highest LRS resistance and the lowest HRS resistance (Fig.R1.2).

Figure R1.2 Cumulative probability distribution of the resistance in HRS and LRS states extracted from DC sweeps at $V_{\text{READ}}=0.1\text{V}$ collected from 200 I-V traces

I would like to see in the article the characteristics (fresh I-V curves) of 5-10 devices without the h-BN, to prove that what they are showing is not produced by the Ti film (which sometimes can easily form TiO₂ and show RS even without an insulating layer).

Thank you for the insightful comment. The process flow adopted for fabricating the devices in this study is as follows,

1. Pattern and deposit bottom electrode (Au)
2. Transfer CVD-grown hBN
3. Pattern and deposit top electrode (Ti/Au)

Both top and bottom electrodes are deposited using electron beam evaporation of metals in a vacuum ($1\text{e-}6$ Torr) environment. During the top electrode deposition, Ti and Au are deposited without breaking the vacuum, thus eliminating Ti's exposure to oxygen. Nevertheless, as rightly pointed out, confirming that the resistive switching in our devices does not originate from an inadvertent TiO₂ layer is critical. Therefore, as suggested, we fabricated and measured the memristor devices without the hBN layer.

Fig.R1.3 shows measured data from 10 devices with different cross-sectional areas ($0.5\times 0.5\mu\text{m}^2 \rightarrow 2\times 2\mu\text{m}^2$). All the devices exhibit ohmic conduction with no conceivable memory window. Moreover, the resistance of these devices is very low (50Ω - 180Ω) compared to the devices with hBN (92% of devices with resistance $> 1\text{M}\Omega$). Therefore, we can ascertain that the resistive switching observed in this study can be attributed to the hBN layer. We have included this figure as Supplementary Fig.8 in the manuscript.

Figure R1.3 DC IV characteristics of Ti/Au devices without the hBN layer. Data collected from devices with varying cross-section areas ($500 \times 500 \text{ nm}^2$ to $2 \times 2 \mu\text{m}^2$).

Regarding the switching mechanism, it is fine for me and properly demonstrated. I would say that the authors could include the EELS data in Figure 2 to make it more powerful. Also, the authors should comment on the recent article demonstrating that non-filamentary RS in h-BN is possible (see <https://www.nature.com/articles/s41586-023-05973-1>)

Thank you for your suggestion to improve the quality of our manuscript. As suggested, we have included the EELS data as Fig.2g in the main text.

The suggested recent article presents interesting results observed in 2D memristors integrated with CMOS transistors. The 1Transistor-1Memristor (1T1M) bitcell exhibited non-filamentary resistive switching with lower transistor V_G (gate voltage) and filamentary switching with higher transistor V_G . This shows controllable soft-breakdown in hBN layers facilitated by the current limiting action of the adjoining transistor. On the contrary, the devices in this study have no current limiting device and exhibit filamentary switching confirmed by ohmic behavior in LRS, positive TCR in LRS, and high LRS currents.

We thank the reviewer for providing this helpful reference highlighting the various possible switching mechanisms in 2D hBN. We have included additional text in the manuscript (lines 167-168 main text) commenting on the reported non-filamentary conduction in hBN.

In the pulsed voltage stresses they should show at least one entire read/write/read/erase/read cycle, so that it can be confirmed that it switches well with cycles. Something like Extended Data Figure 8 in the paper - (<https://www.nature.com/articles/s41586-023-05973-1>)

Thank you for the comment. We have conducted extensive cycling studies on the devices (both 7-8 layer and 3-4 layer hBN). Thick hBN devices show consistent switching for 600 cycles (with $V_{\text{PULSE}} \sim 2.75\text{V}$, $T_{\text{PUL}} \sim 2.7\text{ns}$), and the thin hBN devices exhibit 100 cycles (with $V_{\text{PULSE}} \sim 4\text{V}$, $T_{\text{PUL}} \sim 120\text{ps}$) as seen from Fig.R1.4.

Unfortunately, using our pulse setup, we cannot conduct a measurement similar to Extended Figure 8 of the reference paper. The pulse generator used in this study (PSPL10070A) cannot produce alternating positive and negative pulses in sequence. Therefore, we program the pulse generator every cycle through a computer for the endurance tests. Since the output is measured on a high-frequency scope, two consecutive cycles (separated by milliseconds) cannot be captured due to the limitation of the oscilloscope memory depth (1Gpts).

Figure R1.4 Cycling endurance data for (a) thick devices (7-8 layer hBN) with 2.7ns pulses (b) thin devices (3-4 layer hBN) with 120ps pulses.

The authors should discuss why the endurance is so low. They may show SEM images after the stresses, to see what happens. I guess the wires will melt because the currents are high. Please comment. In fact, I think they could reach higher endurance changing the design of the masks. There is no need to make such long narrow lines, one can reduce the width progressively from the pad as one approaches the Crosspoint section. This reduces current density and slows down electromigration.

We want to answer this question on endurance in the following three sub-sections.

Endurance in 2D memristors

The endurance in 2D memristors has always been a concern, mainly attributed to the ultra-thin switching layer. Transition metal oxide (TMO) based memristors typically have a switching layer thickness ranging between 5nm-50nm [3-4]. On the contrary, 2D memristors can be scaled down to even monolayer thickness. The reduced vertical separation between metal electrodes enhances the electric field, which results in reduced control over the metal ion migration into the switching layer and subsequent filament formation. Moreover, as the thickness of the switching layer reduces, the impact of bottom/top electrode roughness plays a significant role in determining the overall device performance. In the following table, we have listed the reported endurance in 2D memristors along with the thickness of the switching layer.

	Switching layer stack	Thickness (nm)	Endurance cycles
[5]	Au/hBN/Au	0.33	50
[6]	Au/MoS ₂ /Au	0.65	20
[7]	Au/MoS ₂ /Au	0.65	150
[8]	Ag/hBNO _x /Gr	0.9	100
[9]	Au/ReSe ₂ /Au	1	200
[10]	Cu/MoS ₂ /Au	1.4	20
[11]	Ti/hBN/Au	2.4	100
[12]	Ti/hBN/Au	2.4	100
[13]	Ti/hBN/Ni	3.96	100
[14]	Ti/hBN/Au	4.3	50
[15]	Ti/hBN/Au	4.95	250
[16]	Ti/hBN/Gr	5	1000
[17]	Au/MoS ₂ /Au	5	120
[18]	Au/hBN/Au	6	50
[19]	Pd/WS ₂ /Pt	6.45	15
[20]	Ti/hBN/Au	6.6	1200
[21]	Ti/MoTe ₂ /Au	7	600
This work	Ti/hBN/Au	2.64	600
This work	Ti/hBN/Au	1.32	100

From the table, we see that the 2D memristors with switching layer thickness <5nm, typically exhibit <250 cycles endurance. Some memristors [12,16] exhibit about 1000 cycle endurance with >5nm switching layer thickness. In comparison, our devices exhibit on-par or better device performance with 600 cycles for thicker ($T_{\text{hBN}} \sim 2.6\text{nm}$) memristors and 100 for thinner ($T_{\text{hBN}} \sim 1.32\text{nm}$) memristors.

On the other hand, only two research studies report high endurance ($\sim 10^7$) using thicker 2D memristors (>10nm) [22-23]. Many studies report low endurance even with thicker memristors [24-26].

Overall, the 2D material based memristor technology is still in the early stages of exploration. The quality of the 2D material (grain size, uniformity, thickness, defect density etc.,) largely determines the device's performance. Superior control over the CVD growth parameters will enable the fabrication of high-quality memristors with enhanced endurance. With several university and industry research groups working in this direction, we believe that 2D material memristors can outperform oxide-based memristors in the near future.

Role of Current Compliance (CC)

A recent study [27] (suggested by the reviewer in previous comments) reported high endurance of $\sim 10^6$ cycles in the 1T1M (1 Transistor – 1 memristor) bitcell with just $\sim 6.6\text{nm}$ thick switching layer. In this study, the standalone devices without CMOS transistors exhibited only 100 cycles, whereas devices with transistors exhibited a million cycles. This promising result suggests that the adjoining selector device can improve endurance in 2D memristor devices.

The characterization setup utilized in this study cannot support external current compliance (CC) for high-frequency pulse testing. For the frequency range studied in this article, the external CC device (resistor, diode or transistor) must be located within <1cm distance from the DUT (device under test) to be considered as a lumped circuit element. Therefore, the only possible option is to have a CC device fabricated monolithically, like in [27]. Without current compliance, most of our devices get shorted during testing, which cannot be recovered. We expect that having a 1T1M configuration would improve the endurance of our devices.

Current leads

We thank the reviewer for the insightful comment about electromigration in the current leads. However, the devices in our study predominantly fail by getting short due to excess metal ion migration into the ultra-thin switching layer. These excess metal ions form wide filaments which the RESET pulse cannot dissolve, as seen in Fig.R1.5. A similar failure mechanism has also been observed with the DC characterization of the devices. The resistance in this state is <math><100\Omega</math>, which is the range for contact resistance contributed by the current leads.

Figure R1.5 Devices fail by getting shorted after repeated cycling

Electromigration in current leads would have resulted in devices failing in open circuits with high associated resistance. Our devices have bottom and top electrode thicknesses ~70nm, which should lower the probability of electromigration. However, as suggested, we imaged the devices using SEM after stressing the devices with voltage pulses. Fig.R1.6 shows SEM images collected from 4 different devices. The skid marks from the probe landing can be seen on the contact pads. The current leads remain intact without any conceivable electromigration damage.

Figure R1.6 Scanning electron microscope images of the devices after pulse characterization. Skid marks from probe landing on the bond pads are evident.

However, for future studies, we plan to change the mask with a progressively narrowing current leading to lower contact resistance and delayed electromigration, as suggested by the reviewer.

I wonder why the authors don't make TEM images with higher resolution to see more clearly the structure of the devices. It would be good if they could provide something like in this paper: <https://www.nature.com/articles/s41699-022-00341-5>

Thank you for the comment that helps improve the quality of the manuscript. As suggested, we have included TEM images with higher magnification that clearly show the device structure (Fig.R1.7). The hBN film has single-atom defects and few atom wide amorphous regions. These intrinsic defects aid in forming conductive filaments when stressed with high voltages. We have added the high-magnification TEM images as Figure 1e in the main text.

Figure R1.7 TEM images showing single-atom defects (arrows) and few atom wide amorphous regions (dashed circles).

References

1. Yu, S., Gao, B., Dai, H., Sun, B., Liu, L., Liu, X., Han, R., Kang, J. and Yu, B., 2009. Improved uniformity of resistive switching behaviors in HfO₂ thin films with embedded Al layers. *Electrochemical and Solid-State Letters*, 13(2), p.H36.
2. Yuan, B., Liang, X., Zhong, L., Shi, Y., Palumbo, F., Chen, S., Hui, F., Jing, X., Villena, M.A., Jiang, L. and Lanza, M., 2020. 150 nm× 200 nm cross-point hexagonal boron nitride-based memristors. *Advanced Electronic Materials*, 6(12), p.1900115.
3. Shen, Z., Zhao, C., Qi, Y., Xu, W., Liu, Y., Mitrovic, I.Z., Yang, L. and Zhao, C., 2020. Advances of RRAM devices: Resistive switching mechanisms, materials and bionic synaptic application. *Nanomaterials*, 10(8), p.1437.
4. Wong, H.S.P., Lee, H.Y., Yu, S., Chen, Y.S., Wu, Y., Chen, P.S., Lee, B., Chen, F.T. and Tsai, M.J., 2012. Metal-oxide RRAM. *Proceedings of the IEEE*, 100(6), pp.1951-1970.
5. Wu, X., Ge, R., Chen, P.A., Chou, H., Zhang, Z., Zhang, Y., Banerjee, S., Chiang, M.H., Lee, J.C. and Akinwande, D., 2019. Thinnest nonvolatile memory based on monolayer h-BN. *Advanced Materials*, 31(15), p.1806790.
6. Kim, M., Ge, R., Wu, X., Lan, X., Tice, J., Lee, J.C. and Akinwande, D., 2018. Zero-static power radio-frequency switches based on MoS₂ atomistors. *Nature communications*, 9(1), p.2524.

7. Ge, R., Wu, X., Kim, M., Shi, J., Sonde, S., Tao, L., Zhang, Y., Lee, J.C. and Akinwande, D., 2018. Atomistor: nonvolatile resistance switching in atomic sheets of transition metal dichalcogenides. *Nano letters*, 18(1), pp.434-441.
8. Zhao, H., Dong, Z., Tian, H., DiMarzi, D., Han, M.G., Zhang, L., Yan, X., Liu, F., Shen, L., Han, S.J. and Cronin, S., 2017. Atomically thin femtojoule memristive device. *Advanced Materials*, 29(47), p.1703232.
9. Huang, Y., Gu, Y., Wu, X., Ge, R., Chang, Y.F., Wang, X., Zhang, J., Akinwande, D. and Lee, J.C., 2021. ReSe₂-Based RRAM and Circuit-Level Model for Neuromorphic Computing. *Frontiers in Nanotechnology*, 3, p.782836.
10. Xu, R., Jang, H., Lee, M.H., Amanov, D., Cho, Y., Kim, H., Park, S., Shin, H.J. and Ham, D., 2019. Vertical MoS₂ double-layer memristor with electrochemical metallization as an atomic-scale synapse with switching thresholds approaching 100 mV. *Nano letters*, 19(4), pp.2411-2417
11. Shi, Y., Pan, C., Chen, V., Raghavan, N., Pey, K.L., Puglisi, F.M., Pop, E., Wong, H.S. and Lanza, M., 2017, December. Coexistence of volatile and non-volatile resistive switching in 2D h-BN based electronic synapses. In *2017 IEEE International Electron Devices Meeting (IEDM)* (pp. 5-4). IEEE.
12. Shi, Y., Liang, X., Yuan, B., Chen, V., Li, H., Hui, F., Yu, Z., Yuan, F., Pop, E., Wong, H.S.P. and Lanza, M., 2018. Electronic synapses made of layered two-dimensional materials. *Nature Electronics*, 1(8), pp.458-465.
13. Jing, X., Puglisi, F., Akinwande, D. and Lanza, M., 2019. Chemical vapor deposition of hexagonal boron nitride on metal-coated wafers and transfer-free fabrication of resistive switching devices. *2D Materials*, 6(3), p.035021.
14. Wang, C.H., McClellan, C., Shi, Y., Zheng, X., Chen, V., Lanza, M., Pop, E. and Wong, H.S.P., 2018, December. 3D monolithic stacked 1T1R cells using monolayer MoS₂ FET and hBN RRAM fabricated at low (150° C) temperature. In *2018 IEEE International Electron Devices Meeting (IEDM)* (pp. 22-5). IEEE.
15. Yuan, B., Liang, X., Zhong, L., Shi, Y., Palumbo, F., Chen, S., Hui, F., Jing, X., Villena, M.A., Jiang, L. and Lanza, M., 2020. 150 nm × 200 nm cross-point hexagonal boron nitride-based memristors. *Advanced Electronic Materials*, 6(12), p.1900115.
16. Yeh, C.H., Zhang, D., Cao, W. and Banerjee, K., 2020, December. 0.5 T0. 5R-Introducing an Ultra-Compact Memory Cell Enabled by Shared Graphene Edge-Contact and h-BN Insulator. In *2020 IEEE International Electron Devices Meeting (IEDM)* (pp. 12-3). IEEE.
17. Gu, Y., Serna, M.I., Mohan, S., Londoño-Calderon, A., Ahmed, T., Huang, Y., Lee, J., Walia, S., Pettes, M.T., Liechti, K.M. and Akinwande, D., 2022. Sulfurization Engineering of One-Step Low-Temperature MoS₂ and WS₂ Thin Films for Memristor Device Applications. *Advanced Electronic Materials*, 8(2), p.2100515.
18. Shen, Y., Zheng, W., Zhu, K., Xiao, Y., Wen, C., Liu, Y., Jing, X. and Lanza, M., 2021. Variability and Yield in h-BN-Based Memristive Circuits: The Role of Each Type of Defect. *Advanced Materials*, 33(41), p.2103656.
19. Yan, X., Zhao, Q., Chen, A.P., Zhao, J., Zhou, Z., Wang, J., Wang, H., Zhang, L., Li, X., Xiao, Z. and Wang, K., 2019. Vacancy-induced synaptic behavior in 2D WS₂ nanosheet-based memristor for low-power neuromorphic computing. *Small*, 15(24), p.1901423
20. Zhuang, P., Lin, W., Ahn, J., Catalano, M., Chou, H., Roy, A., Quevedo-Lopez, M., Colombo, L., Cai, W. and Banerjee, S.K., 2020. Nonpolar resistive switching of multilayer-hBN-based memories. *Advanced Electronic Materials*, 6(1), p.1900979.
21. Zhang, F., Zhang, H., Shrestha, P.R., Zhu, Y., Maize, K., Krylyuk, S., Shakouri, A., Campbell, J.P., Cheung, K.P., Bendersky, L.A. and Davydov, A.V., 2018, December. An ultra-fast multi-level MoTe₂-based RRAM. In *2018 IEEE International Electron Devices Meeting (IEDM)* (pp. 22-7). IEEE.
22. Tang, B., Veluri, H., Li, Y., Yu, Z.G., Waqar, M., Leong, J.F., Sivan, M., Zamburg, E., Zhang, Y.W., Wang, J. and Thean, A.V., 2022. Wafer-scale solution-processed 2D material analog resistive memory array for memory-based computing. *Nature Communications*, 13(1), p.3037.

23. Wang, M., Cai, S., Pan, C., Wang, C., Lian, X., Zhuo, Y., Xu, K., Cao, T., Pan, X., Wang, B. and Liang, S.J., 2018. Robust memristors based on layered two-dimensional materials. *Nature Electronics*, 1(2), pp.130-136.
24. Braun, D., Lukas, S., Völkel, L., Hartwig, O., Prechtel, M., Belete, M., Kataria, S., Wahlbrink, T., Daus, A., Duesberg, G.S. and Lemme, M.C., 2022, June. Non-Volatile Resistive Switching in PtSe₂-Based Crosspoint Memristors. In *2022 Device Research Conference (DRC)* (pp. 1-2). IEEE.
25. Jeong, H.Y., Kim, J.Y., Kim, J.W., Hwang, J.O., Kim, J.E., Lee, J.Y., Yoon, T.H., Cho, B.J., Kim, S.O., Ruoff, R.S. and Choi, S.Y., 2010. Graphene oxide thin films for flexible nonvolatile memory applications. *Nano letters*, 10(11), pp.4381-4386.
26. Wang, Y., Yang, J., Wang, Z., Chen, J., Yang, Q., Lv, Z., Zhou, Y., Zhai, Y., Li, Z. and Han, S.T., 2019. Near-infrared annihilation of conductive filaments in quasiplane MoSe₂/Bi₂Se₃ nanosheets for mimicking heterosynaptic plasticity. *Small*, 15(7), p.1805431.
27. Zhu, K., Pazos, S., Aguirre, F., Shen, Y., Yuan, Y., Zheng, W., Alharbi, O., Villena, M.A., Fang, B., Li, X. and Milozzi, A., 2023. Hybrid 2D/CMOS microchips for memristive applications. *Nature*, pp.1-3.

Reviewer 2

In the introduction, the authors noted that there has been relatively little attention devoted to studying the switching dynamics of memristors. However, numerous studies have explored the dynamic mechanisms in both metal-oxide-based and hBN-based memristors. For instance, TEM has been used to observe the electrochemical dynamics of nanoscale metal conductive filaments in metal-oxide-based memristors, and a model has been established to consider the applied electric field, metal ion diffusion, and filament shape based on dynamic metal cluster nucleation and growth. [1,2] In the case of 2D hBN-based memristors, the dynamic analysis of nano-conductive filaments has also been investigated, including the effect of temperature. [3,4] Additionally, a Monte Carlo simulation has been shown to simulate filament growth and erase dynamics. [5]

Thank you for providing valuable references that we missed during our literature survey. Through the statement, “little attention devoted to studying the switching dynamics of memristors,” we meant to say that studying the switching dynamics through transient switching characteristics was less explored. However, we agree that the statement is ambiguous, and considerable research effort has been dedicated to studying the switching dynamics in memristors. Therefore, to avoid confusion, the sentence has been removed from the manuscript, and the introduction has been updated as suggested by another reviewer. We have included the suggested references for completeness.

In Figure 2a, have the initial currents measured from devices with different areas been plotted together? If not, the authors should clearly state the device area that used in this figure. If initial currents have been plotted together, the authors should include a legend specifying the device area, as this parameter is critical in determining the high-resistance state (HRS) initial current. A similar issue is observed in Supplementary Figure 7a, where the authors plot the correlation between the forming voltage and the initial current. In this case, the authors should keep the device area constant.

Thank you for the suggestion. The plot (Fig.2a in original manuscript) includes the initial resistance measured from devices with different cross-section areas. We agree that the device area determines the initial resistance and should have been plotted separately. As suggested, we have updated the figure by segregating the data points based on the device area and plotting the graphs separately with a corresponding legend (Figure R2.1). It is evident that the forming voltage increases with initial resistance for all device dimensions. Therefore, a generic relationship can be established between initial resistance and forming voltage for the 2D hBN based memristor devices.

Figure R2.1 Forming Voltage vs Initial Resistance for devices with contact dimensions (a) 0.5um (b) 0.75um (c) 1um (d) 1.5um (e) 2um

Similarly, we have updated Supplementary Figure 7a by segregating the data points based on device dimensions (Figure R2.2). As expected, the initial resistance distribution moves to lower values as the device's cross-sectional area increases.

Figure R2.2 Cumulative distribution of Initial Resistance for devices with varying contact dimensions

Both these figures have been included as Supplementary Fig.7 in the manuscript.

In Fig. 2d, the device cycle-to-cycle variation is much higher than other hBN-based memristors.[6] The range of set voltages is wide, spanning from 1 to 3 V, creating significant difficulties in selecting memristor devices for cross bar arrays. Furthermore, device-to-device variation exacerbates this issue. To understand the cause of this considerable variation, the authors should conduct further analysis.

Thank you for the insightful comment. Fig.2c (updated manuscript number) plots the SET/RESET voltages from 200 IV traces. These traces include multiple cycles collected from different devices. The overall variation of the SET voltage spans between 1V to 3V is not ideal for large cross-bar array applications. To understand the source of this variation, we segregated the data into cycle-to-cycle and device-to-device components. Qualitatively, we observed that the cycle-to-cycle variation is lower than the device-to-device variation. This is evident in Figure. R2.3, where DC IV characteristics collected from four devices are presented. Although the mean SET voltage varies considerably (between 1.2V and 2.54V), the distributions are tightly bound around the mean value with a worst-case standard deviation of 0.25V. This indicates that the device-to-device variation is the dominant cause of variation for these memristors.

Figure R2.3 DC IV characteristics collected from four different memristor devices.

We speculate that the electrode surface roughness contributes to the higher device-to-device variation among these devices. Evaporated metal electrodes typically have a surface roughness of about 1-2nm [11-13]. The effect of this roughness on the switching characteristics should be lower in memristors with thicker switching layer (>5nm). This could potentially explain the reduced variation observed in the memristors studied in [6], where 18-layer hBN (~6.6nm thick) was utilized to fabricate the devices. However, as the switching layer thickness scales down, it is easier for filaments to propagate vertically and form around weak spots with bumps/hillocks. Therefore, the surface roughness of the electrodes will largely determine the location and voltage of the breakdown. These variations can be reduced with industry-standard tools that can achieve lower surface roughness, which is difficult to achieve with university equipment.

In the study on dynamic filament formation (see Fig. 4 and Supplementary Fig. 10), the authors employed a conventional model that takes into account the effects of temperature, the heat equation, and the Arrhenius relationship for filament growth. It should be noted, however, that this model was originally developed for oxide memristors. [7,8] As such, it is unclear how the model in this work differs from the one used for metal oxide, and how it specifically contributes to understanding the filament growth dynamics in hBN.

Thank you for the comment. We agree that the Arrhenius model was originally developed for oxide memristors. The corresponding references have been duly cited in the manuscript (references 55-57) and the supplementary information (references S18-S19). This universal phenomenological model satisfactorily captures the growth dynamics in memristors such as OxRRAMs (Oxide RRAM) and CBRAMs (conductive bridge RAM). Therefore, in this study, we employ the same model with modifications to capture the behavior of our 2D memristors under ultra-fast voltage pulse stresses.

The models presented in [7,8] consider filament formation and dissolution as independent events i.e., during a SET process, all the equations only model filament expansion. However, in reality, formation and dissolution proceed simultaneously (determined by voltage and temperature), albeit at different rates. For example, as the filament begins to widen, the temperature generated in the filament increases, which promotes outward diffusion of the metal ions from the filament. Since the memristors in this study have an ultra-thin switching layer (<2.5nm), very low resistance filaments (200Ω -500Ω) are formed during the SET process. These low-resistance filaments generate high filament temperatures that promote dissolution resulting in filament narrowing. This phenomenon of filament narrowing was evidently observed in our devices (Fig.R2.4). For this study, we measured the resistance of the device before (R_{INIT}), after (R_{FIN}), and during the pulse (R_{PUL}). The resistance distributions of R_{INIT} , R_{PUL} , and R_{FIN} (Fig.R2.4b) show that the memristor resistance reduces during the SET pulse and again increases after the pulse suggesting a reduction in filament cross-section.

Figure R2.4 (a) Voltage and current pulse waveforms during a SET operation. (b) Resistance distribution of R_{INIT} , R_{PUL} , R_{FIN} . The R_{PUL} was estimated as the resistance value calculated at the maximum current magnitude during the pulse. The resistance value is calculated at the maximum current magnitude during the pulse (R_{PUL}) and immediately before (R_{INIT}) and after (R_{FIN}) the pulse via applying a small DC bias $V=100mV$.

The Arrhenius models (Eq.R2.1) [7,8] can only capture filament expansion during the SET process and vice versa during the RESET process. Therefore, the final resistance estimated by these models will need to be revised.

$$\frac{d\Phi}{dt} = Ae^{\left(-\frac{E_{a0}-\alpha qV}{kT_{CF}}\right)} \quad \text{----- (R2.1)}$$

To account for filament narrowing, we include a second term that captures the dynamics of filament dissolution (Eq.R2.2).

$$\frac{d\Phi}{dt} = A_1e^{\left(-\frac{E_{a0}-\alpha qV}{kT_{CF}}\right)} - A_2e^{\left(-\frac{E_a}{kT_{CF}}\right)} \quad \text{----- (R2.2)}$$

Note that the activation energy E_a required for metal ion out-diffusion from the filament differs from activation energy E_{a0} for filament formation (bond breaking and ion hopping). As the voltage pulse ramps up, the first term dominates, and the filament expands. After reaching a certain filament width the higher current flowing through the device generates higher temperature. Here, the filament growth rate is balanced by the filament dissolution rate, and the filament diameter saturates. However, during the voltage pulse falling transition, the growth rate (determined by V , T_{CF}) decreases faster than the dissolution rate (determined only by T_{CF}), thereby resulting in the narrowing of the filament.

Figure R2.5 (a) Voltage and current pulse waveforms during a SET operation. The model current closely traces the measured current. Calculated time evolution of (b) filament diameter (c) memristor resistance.

A self-consistent electro-thermal solver using the proposed model clearly captures the filament narrowing (Fig.R2.5b,c). The filament expands to ~9nm in diameter and narrows to ~6.5nm after pulse termination. The memristor resistance change closely matches the resistance values presented in Fig.R2.4b. We believe that this joule heating driven filament narrowing will be observed in other 2D memristors as well, and therefore the proposed equations can be used to model them.

We have added additional text in the supplementary and the main text to highlight how our model differs from the previously published models [7,8].

The authors need to provide a clear explanation of the fundamental reason for the ultra-fast switching behavior observed in their memristor, as compared to other fast-switching memristors discussed in previous studies.[9,10] To achieve this, the authors should provide a detailed explanation in terms of their model and material properties.

Thank you for the valuable and insightful comment. The memristors in previous studies [9,10] switch faster due to their unique switching mechanism, which is completely different from traditional Oxide RRAMs or CBRAMs. In [9], the memristor has Pt nanoparticles dispersed across the switching layer, effectively reducing the overall separation between the electrodes, which results in ultra-fast switching. On the other hand, the devices in [10] have a conducting channel in both ON and OFF states with different chemical compositions and therefore switch faster, whereas the switching mechanism in our 2D memristor is based on metal ion diffusion into the switching layer. The applied electric field facilitates the vertical propagation of the filament around weak spots in the dielectric. This initially formed narrow filament connects the electrodes and lowers the resistance of the memristor. The current flowing through this filament generates temperature through the Joule heating phenomenon. The temperature generated in the filament lowers the energy barrier for ion release from the electrode and further propels the filament growth. To understand the fundamental reason for the ultra-fast switching in our devices, we perform transient thermal analysis using finite element based physics solvers (COMSOL). To summarize the results, the characteristic properties aiding the ultra-fast switching in 2D hBN memristors are:

1. Ultra-thin switching layer
2. High thermal conductivity of 2D hBN layers

Figure R2.6 (a) Electric field vs dielectric thickness for V=1V. (b) Snapshot of the COMSOL model showing the metallic filament and electrodes. The temperature profile shows the highest temperature at the center of the filament. (c) Temperature reaching the interface vs dielectric thickness (d) time taken for the interface to reach 400K, 500K, 600K as a function of dielectric thickness.

Ultra-thin switching layer

Oxide memristors (commercial or university demonstrations) typically have a thicker switching layer (5nm-20nm) in comparison to the memristors in this study (< 2.4nm). The thinner switching layer significantly increases the local electric field across the dielectric, which promotes faster resistance switching. Fig.R2.6a shows the electric field increase inside the switching layer as the dielectric thickness reduces.

From the Arrhenius model (Eq.R2.1,2), we see that the filament growth dynamics depend on the temperature in addition to the applied voltage. The critical physical process that determines the switching speed of the device is the metal ion release [7] which is determined by the Ti/hBN interface temperature (T_{INT}). Note that this temperature is different from the filament temperature (T_{CF}). The current passing through the device generates temperature in the filament (T_{CF}) through joule heating. This temperature is highest at the center of the filament (vertically) and lowers towards the electrodes, as seen in Fig.R2.6b. This nature of the temperature profile has been reported in several previous studies [14-16]. Most studies approximate $T_{CF} \sim T_{INT}$ for simplicity in analytical modeling.

Using the COMSOL model, we study the T_{INT} dependence on dielectric thickness, as shown in Fig.R2.6c. It is evident that as the dielectric thickness reduces, steady-state T_{INT} increases. In addition, Fig.R2.6d plots the time taken for temperature build-up at the interface as a function of dielectric thickness. We plot the time taken to reach 400K, 500K, and 600K for comparison. The T_{INT} for thinner dielectric (such as the devices in this study) reaches 600K in less than 1ps, whereas thicker dielectric ~5nm takes about 40ps. The faster temperature build-up at the interface promotes faster metal ion release, which creates a positive feedback loop: higher $T_{INT} \rightarrow$ lower $R \rightarrow$ higher T_{INT} . This positive feedback loop promotes faster switching in devices with a thinner dielectric layer.

Figure R2.7 (a) Snapshot of COMSOL model showing three points of interest A,B,C. Transient interface temperature at (b) point A (c) point B (d) point C.

High thermal conductivity of 2D hBN

Based on the previous discussion, the obvious subsequent question would be - “Will the oxide RRAMs with thinner dielectric layer switch as fast as the 2D memristors?” To answer this question, we simulated two devices – one with an HfO_x layer and the other with 2D hBN, with identical switching layer thicknesses (2.5nm).

In addition to thickness, another physical property that determines the switching speed of the devices is the thermal conductivity of the switching layer. The in-plane thermal conductivity of hBN ($k \sim 100\text{-}200 \text{ W/mK}$) [17-19] is significantly higher than oxides ($k \sim 0.5\text{-}2 \text{ W/mK}$) typically used for memristors (HfO_2 , Al_2O_3 , TaO_x) [20-22]. On the other hand, the out-of-plane thermal conductivity is comparable (5-10 mW/K) to the oxides. The high thermal conductivity quickly spreads the heat from the filament to the surroundings, which heats a larger cross-section of the interface. Fig.R2.7a shows three points on the interface at 2nm, 4nm, and 6nm away from the filament. Temperature transients at these points (Fig.R2.7b,c,d) highlight this effect. Evidently, T_{INT} increases rapidly for hBN compared to HfO_x . This effect is especially more pronounced as we move away from the filament (points B, C). Since a larger cross-section of the interface is heated, the probability of ion release increases, which results in faster filament expansion. Therefore, 2D materials that typically have higher thermal conductivity promote faster switching compared to oxide RRAMs with identical layer thickness.

Overall, the 2D memristors benefit from having an ultra-thin switching layer as well as high thermal conductivity compared to TMO memristors. Fig.R2.8 shows the T_{INT} for hBN devices with 2.5nm thickness and HfO_x devices with 5nm thickness (typical thickness for TMO memristors). Clearly, the temperature rises rapidly for the hBN devices resulting in ultra-fast switching.

This explanation for ultra-fast switching observed in 2D memristors has been added as Supplementary Figure.16 in the manuscript.

Figure R2.8 Average transient interface temperature for hBN with thickness=2.5nm and HfO_x with thickness=5nm

It would be beneficial if the authors could compare the fitted filament diameter obtained from their model with experimental results obtained from techniques such as C-AFM or HRTEM, which can directly measure the filament diameter.

Thank you for the comment. Obtaining the filament diameter through HRTEM is difficult as the probability that the FIB cut exactly passes through the filament is low; unfortunately, even after extensive searching across the FIB cuts, we did not observe filaments in our experiments. Moreover, all the elements involved in our devices (B,N,Ti) have relatively smaller atomic weights that do not produce discernible differences in contrast. Although some discontinuities in the layered hBN film are observed, these can, at best, be characterized as some defects or amorphous regions rather than as conductive filaments.

Figure R2.9 TEM images showing single-atom defects (arrows) and few atom wide amorphous regions (dashed circles).

Even if a filament existed in the FIB cut (typical depth $\sim 100\text{nm}$), it may not be visible in the TEM image unless its dimension is comparable to the depth. Since the 2D memristors have an ultra-thin switching layer, narrow filaments can carry large currents compared to the thicker TMO memristors. Therefore, such large filament cross-section typically are not created in 2D memristors (typical filament dimension $\sim 10\text{-}20\text{nm}$ [23]). The low contrast of Ti combined with the small dimension makes filament imaging in 2D memristors very challenging. All the previous studies (2D materials or otherwise) that have successfully imaged filaments through HRTEM have Ag, Au, or some other heavy elements that can produce sufficient contrast [1-2,23-24]. Other noted studies [25-26] in the field using hBN and Ti material system report defects/amorphous regions similar to Fig.R2.9.

At this time, unfortunately, the conductive AFM at our facility is not functional and would need several months before replacement parts can arrive.

However, our model's filament diameter matches closely with the experimental filament diameter ($\sim 10\text{nm}$) obtained through HRTEM and CAFM published in previous study on 2D hBN memristors [23].

The authors should discuss more metrics or guidelines for measuring the pulse characteristics of their memristor. In Figures 4 and 5, the authors have presented statistical analysis of the transient pulse response and established correlations between switching ratio, switching time, and switching energy. It shows that the switching energy and switching time is highly related to the switching ratio before/after pulse. To better understand the performance of their ultra-fast switching device, the authors should discuss key performance metrics for fast-switching devices and the differences between normal endurance (with pulse widths of tens of microseconds) and ultra-fast pulse endurance (with sub-nanosecond pulses).

Thank you for the valuable comment. Based on our experience, we have listed the following guidelines and metrics for testing memristors with ultra-fast pulses.

Signal Reflections

Characterization of the memristors in the sub-nanosecond regime requires the fast pulse to be delivered through a transmission line setup using specialized RF cables with matching impedance (50Ω). The pulse delivered to the device-under-test (DUT) needs to be monitored for further analysis. This can be achieved by sampling the input transmission line through a pick-off tee. In addition to using RF cables, the transmission line needs to be properly terminated with a 50Ω resistor, placed as close as possible to the DUT, ideally on the probe tips, as shown in Fig.R2.10. This termination resistor minimizes reflections and ensures maximum power transfer to the DUT.

Figure R2.10 Test setup schematic showing the 50Ω termination placed near the probe tips.

Figure R2.11 Transient waveform for (a) SET pulse with 50Ω resistor (b) SET pulse without 50Ω resistor (c) RESET pulse with 50Ω resistor (d) RESET pulse without 50Ω resistor.

The 50Ω termination resistor effectively suppresses reflections, as seen in Fig.R2.11a,c for SET and REST pulses, respectively. Whereas the reflections are significant when the 50Ω resistor was not soldered onto the probe tips, as seen in Fig.R2.11b,d. Such reflections would result in random programming of the memristor. Therefore, all the experiments presented in this work were conducted with the 50Ω termination on the probe tips ensuring minimal reflected power.

Test setup parasitics

Figure R2.12 Applied voltage pulse (red trace) and measured current (blue trace) across a discrete surface mount resistor with the resistance (a) 1kΩ (b) 500Ω (c) 200Ω. (d) I-V plots of the pulses are shown in (a),(b),(c).

The test setup needs to be vetted thoroughly for parasitics, reflections, and noise sources before proceeding to test memristors. If noise appears in the voltage pulses, each segment of the transmission line needs to be tested in isolation with a 50Ω termination. In addition, any unintended parasitic capacitance in the setup will produce a delayed current response that could be mistaken as resistive switching. The parasitics in the setup can be identified by applying voltage pulses to standalone resistors (SMD or other) and verifying the response. For example, we observed hysteresis when voltage pulses were applied to the memristors. To ensure that the hysteresis originated from memristors and not system parasitics, we tested the response of resistors. Fig.R2.12 shows that no significant hysteretic window is observed, suggesting that the hysteresis can be attributed to the resistive switching of the memristor.

Sanity checks for confidence

In our experiments, we observed that the resistance of the memristor during the SET pulse is lower than the resistance after the pulse. As explained in the methods section, the resistance during the pulse is directly measured by the oscilloscope, whereas the resistance before/after the pulse is measured through a current amplifier in the branched DC path. Therefore, any additional resistance in the DC path can increase the resistance after the pulse. To confirm that the DC path does not have any parasitic resistance, we applied the voltage pulse to a standalone resistor and measured the resistance. Fig.R2.13 shows that the resistance during and after the pulse measured close to 1.2kΩ. Therefore, the resistance increase observed after the pulse in Fig.5c (main text) is indeed a characteristic feature of the memristors associated with joule heating. Such sanity checks at every stage are necessary to ensure the correct interpretation of the experimental observations.

Figure R2.13 SET pulse applied to discrete resistor, (a) Voltage and current waveforms (b) DC current measured before and after the pulse

Switching time

Switching time is the primary performance metric for fast-switching memristors intended for high-frequency applications. The switching time of a memristor can be characterized by measuring the current response to the voltage pulse. When characterizing the switching time, some reports do not consider the voltage pulse rise time. In addition, few studies define switching time as the duration required to reach 50% of the final resistance level. Although such characterization techniques provide an attractive switching time estimate, we believe that this does not reflect a realistic scenario. For example, some signals switch faster up to 50% resistance level but may slow down later, which will not be captured by 50% definition. Therefore, we recommend including the voltage pulse rise time as well as defining the switching speed as the time taken for the current signal to settle to ~90% of its final resistance state.

Endurance Cycling

Pulse testing in the sub-nanosecond regime is tricky (test setup and analysis) and complicated. Therefore, most studies typically characterize the switching speed based on a single experiment. Characterizing the memristor speed based on a single cycle is unreliable and potentially misleading. It is difficult to ascertain that the observed fast response is associated with a reversible switching process. For example, a device shorted (failed) by the fast pulse will produce a current signature similar to SET. Likewise, a device whose current leads are damaged due to excessive current (electromigration) will produce a signature similar to RESET. Therefore, the switching speed of the device should only be characterized by considering subsequent cycling data. In this regard, only two research articles show repeatable switching with 10 cycles [9] and 20 cycles [27] using ultra-short voltage pulses. The fact that many studies do not show cycling data may suggest that those devices do not switch consistently. In addition, cycling experiments provide statistical data on the switching time of the memristors, which is crucial to understand the variations in the switching time.

Cycling with wider pulses

To understand the impact of pulse width on endurance cycling, we measured our devices with 100ns voltage pulses (Fig.R2.14a). We observe that the devices cycle for about 60-80 cycles (Fig.R2.14b) before getting shorted. Although it may seem like the cycling performance of the device degrades with wider pulses, such an interpretation would be premature. Rather than the pulse width, the endurance in our experiments is limited by the test setup and device configuration. The primary concern here (both for narrow and wide pulses) is the absence of a current limiting

mechanism which results in excess metal ion diffusion into the switching layer. The excess metal ions form wide filaments in the ultra-thin switching layer, which the RESET pulse cannot dissolve. A similar failure mechanism has also been observed with the DC characterization of the devices. Therefore, it is critical to limit the metal ion diffusion into the memristor either through a barrier layer or an external current-limiting device. A very recent study [28] reported high endurance of $\sim 10^6$ cycles in the 1T1M (1transistor-1memristor) bitcell hBN switching layer. In that study, the standalone devices without CMOS transistors exhibited only 100 cycles, whereas devices with transistors exhibited a million cycles. This promising result suggests that the current limiting mechanism is critical to improving endurance in 2D memristor devices.

Figure R2.14 (a) Pulse voltage and current waveforms (b) Cycling data with 100ns voltage pulses

The authors demonstrate 600-cycle endurance with ultra-fast pulses, which is higher than other fast memristors' endurance results. However, it is important to note that other memristors can achieve endurance up to 108 cycles with pulse widths of microseconds. Therefore, to prove the stability of their device, the authors should provide endurance results with a higher cycle number.

Thank you for the comment. We want to answer this question on endurance through the following sub-sections,

Endurance in 2D memristors

The endurance in 2D memristors has been a concern, mainly attributed to the ultra-thin switching layer. Transition metal oxide (TMO) based memristors typically have a switching layer thickness ranging between 5nm-20nm [29-31]. On the contrary, 2D memristors typically have ultra-thin switching layers, sometimes even scaled down to monolayer thickness. The reduced vertical separation between metal electrodes enhances the electric field, which results in reduced control over the metal ion migration into the switching layer and subsequent filament formation. Moreover, as the thickness of the switching layer reduces, the impact of bottom/top electrode roughness plays a significant role in determining the overall device performance. In the following table, we have listed the reported endurance in 2D memristors along with the thickness of the switching layer.

	Switching layer stack	Thickness (nm)	Cycles endurance
[32]	Au/hBN/Au	0.33	50
[33]	Au/MoS ₂ /Au	0.65	20
[34]	Au/MoS ₂ /Au	0.65	150
[35]	Ag/hBNO _x /Gr	0.9	100
[36]	Au/ReSe ₂ /Au	1	200
[37]	Cu/MoS ₂ /Au	1.4	20
[38]	Ti/hBN/Au	2.4	100
[39]	Ti/hBN/Au	2.4	100
[40]	Ti/hBN/Ni	3.96	100
[41]	Ti/hBN/Au	4.3	50
[42]	Ti/hBN/Au	4.95	250
[43]	Ti/hBN/Gr	5	1000
[44]	Au/MoS ₂ /Au	5	120
[45]	Au/hBN/Au	6	50
[46]	Pd/WS ₂ /Pt	6.45	15
[47]	Ti/hBN/Au	6.6	1200
[48]	Ti/MoTe ₂ /Au	7	600
This work	Ti/hBN/Au	2.64	600
This work	Ti/hBN/Au	1.32	100

From the table, we see that the 2D memristors with switching layer thickness <5nm, typically exhibit <250 cycles endurance. Some memristors [43,47] exhibit about 1000 cycle endurance with >5nm switching layer thickness. Our devices exhibit on-par performance with 600 cycles for thicker ($T_{\text{hBN}} \sim 2.64\text{nm}$) memristors and 100 for thinner ($T_{\text{hBN}} \sim 1.32\text{nm}$) memristors.

On the other hand, only two research studies report high endurance ($\sim 10^7$) using thicker 2D memristors [49-50]. Many studies report low endurance even with thicker 2D memristors [51-53].

Role of Current Compliance (CC)

A recent study [28] (suggested by the reviewer in previous comments) reported high endurance of $\sim 10^6$ cycles in the 1T1M (1transistor-1memristor) bitcell with just $\sim 6.6\text{nm}$ thick switching layer. In that study, the standalone devices without CMOS transistors exhibited only 100 cycles, whereas devices with transistors exhibited a million cycles. This promising result suggests that the adjoining selector device can improve endurance in 2D memristor devices.

The characterization setup utilized in this study cannot support external current compliance (CC) for high-frequency pulse testing. For the frequency range studied in this article, the external CC device (resistor, diode or transistor) must be located within <1cm distance from the DUT (device under test) to be considered as a lumped circuit element. Therefore, the only possible option is to have a CC device fabricated monolithically, like in [28]. Without current compliance, most of our devices get shorted during testing, which cannot be recovered.

We have tested multiple devices but unfortunately could not get better endurance cycling than 600 cycles with the 2.7ns pulses. However, we believe that having a 1T1M configuration would improve the endurance of our devices as well.

There are some typos in the manuscript. (Line 81, the SET should be transition to lower resistance instead of lower conductance)

Thank you for pointing out the typo. We have corrected the sentence. We have thoroughly scanned the manuscript for typos.

The image quality is poor in the manuscript, some of the images cannot be viewed clearly. (Figure 2g, 2h)

We have updated the figures 2g,2h with better quality images and increased the plot texts for improved readability.

Scale bar labeling is confusing. In Supplementary Fig. 3a, the sample electrode width seems to be larger than 2 μm . However, the maximum electrode width is 2 μm in this work.

Thank you for pointing that out, this would have missed our attention. There was a mistake in generating the scale bar. We have updated the figure with the correct scale bar.

References

- [1] Yang, Y., Gao, P., Li, L., Pan, X., Tappertzhofen, S., Choi, S., Waser, R., Valov, I. and Lu, W.D., 2014. Electrochemical dynamics of nanoscale metallic inclusions in dielectrics. *Nature communications*, 5(1), p.4232.
- [2] Yang, Y., Gao, P., Gaba, S., Chang, T., Pan, X. and Lu, W., 2012. Observation of conducting filament growth in nanoscale resistive memories. *Nature communications*, 3(1), p.732.
- [3] Lanza, M., Palumbo, F., Shi, Y., Aguirre, F., Boyeras, S., Yuan, B., Yalon, E., Moreno, E., Wu, T. and Roldan, J.B., 2022. Temperature of conductive nanofilaments in hexagonal boron nitride based memristors showing threshold resistive switching. *Advanced Electronic Materials*, 8(8), p.2100580.
- [4] Zhu, K., Liang, X., Yuan, B., Villena, M.A., Wen, C., Wang, T., Chen, S., Hui, F., Shi, Y. and Lanza, M., 2019. Graphene–boron nitride–graphene cross-point memristors with three stable resistive states. *ACS applied materials & interfaces*, 11(41), pp.37999-38005.
- [5] Qian, F., Chen, R.S., Wang, R., Wang, J., Xie, P., Mao, J.Y., Lv, Z., Ye, S., Yang, J.Q., Wang, Z. and Zhou, Y., 2022. A leaky integrate-and-fire neuron based on hexagonal boron nitride (h-BN) monocrystalline memristor. *IEEE Transactions on Electron Devices*, 69(11), pp.6049-6056.
- [6] Chen, S., Mahmoodi, M.R., Shi, Y., Mahata, C., Yuan, B., Liang, X., Wen, C., Hui, F., Akinwande, D., Strukov, D.B. and Lanza, M., 2020. Wafer-scale integration of two-dimensional materials in high-density memristive crossbar arrays for artificial neural networks. *Nature Electronics*, 3(10), pp.638-645.

- [7] Huang, P., Liu, X.Y., Chen, B., Li, H.T., Wang, Y.J., Deng, Y.X., Wei, K.L., Zeng, L., Gao, B., Du, G. and Zhang, X., 2013. A physics-based compact model of metal-oxide-based RRAM DC and AC operations. *IEEE transactions on electron devices*, 60(12), pp.4090-4097.
- [8] Ielmini, D., 2011. Modeling the universal set/reset characteristics of bipolar RRAM by field-and temperature-driven filament growth. *IEEE Transactions on Electron Devices*, 58(12), pp.4309-4317.
- [9] Choi, B.J., Torrezan, A.C., Norris, K.J., Miao, F., Strachan, J.P., Zhang, M.X., Ohlberg, D.A., Kobayashi, N.P., Yang, J.J. and Williams, R.S., 2013. Electrical performance and scalability of Pt dispersed SiO₂ nanometallic resistance switch. *Nano letters*, 13(7), pp.3213-3217.
- [10] Choi, B.J., Torrezan, A.C., Strachan, J.P., Kotula, P.G., Lohn, A.J., Marinella, M.J., Li, Z., Williams, R.S. and Yang, J.J., 2016. High-speed and low-energy nitride memristors. *Advanced Functional Materials*, 26(29), pp.5290-5296.
- [11] Chen, G. and Hui, P., 1999. Thermal conductivities of evaporated gold films on silicon and glass. *Applied physics letters*, 74(20), pp.2942-2944.
- [12] Yang, Z., Liu, C., Gao, Y., Wang, J. and Yang, W., 2016. Influence of surface roughness on surface plasmon resonance phenomenon of gold film. *Chinese Optics Letters*, 14(4), p.042401.
- [13] Stenzel, O., Wilbrandt, S., Stempfhuber, S., Gäbler, D. and Wolleb, S.J., 2019. Spectrophotometric characterization of thin copper and gold films prepared by electron beam evaporation: thickness dependence of the Drude damping parameter. *Coatings*, 9(3), p.181.
- [14] Niraula, D. and Karpov, V.G., 2017. Heat transfer in filamentary RRAM devices. *IEEE Transactions on Electron Devices*, 64(10), pp.4106-4113.
- [15] Ielmini, D., Nardi, F. and Cagli, C., 2011. Physical models of size-dependent nanofilament formation and rupture in NiO resistive switching memories. *Nanotechnology*, 22(25), p.254022.
- [16] Larentis, S., Nardi, F., Balatti, S., Gilmer, D.C. and Ielmini, D., 2012. Resistive switching by voltage-driven ion migration in bipolar RRAM—Part II: Modeling. *IEEE Transactions on Electron Devices*, 59(9), pp.2468-2475.
- [17] Jo, I., Pettes, M.T., Kim, J., Watanabe, K., Taniguchi, T., Yao, Z. and Shi, L., 2013. Thermal conductivity and phonon transport in suspended few-layer hexagonal boron nitride. *Nano letters*, 13(2), pp.550-554.
- [18] Alam, M.T., Bresnehan, M.S., Robinson, J.A. and Haque, M.A., 2014. Thermal conductivity of ultra-thin chemical vapor deposited hexagonal boron nitride films. *Applied Physics Letters*, 104(1).
- [19] Jana, M. and Singh, R.N., 2018. Progress in CVD synthesis of layered hexagonal boron nitride with tunable properties and their applications. *International Materials Reviews*, 63(3), pp.162-203.
- [20] Panzer, M.A., Shandalov, M., Rowlette, J.A., Oshima, Y., Chen, Y.W., McIntyre, P.C. and Goodson, K.E., 2009. Thermal properties of ultrathin hafnium oxide gate dielectric films. *IEEE Electron Device Letters*, 30(12), pp.1269-1271.
- [21] Cappella, A., Battaglia, J.L., Schick, V., Kusiak, A., Lamperti, A., Wiemer, C. and Hay, B., 2013. High Temperature Thermal Conductivity of Amorphous Al₂O₃ Thin Films Grown by Low Temperature ALD. *Advanced Engineering Materials*, 15(11), pp.1046-1050.
- [22] Landon, C.D., Wilke, R.H., Brumbach, M.T., Brennecke, G.L., Blea-Kirby, M., Ihlefeld, J.F., Marinella, M.J. and Beechem, T.E., 2015. Thermal transport in tantalum oxide films for memristive applications. *Applied Physics Letters*, 107(2).
- [23] Shi, Y., Liang, X., Yuan, B., Chen, V., Li, H., Hui, F., Yu, Z., Yuan, F., Pop, E., Wong, H.S.P. and Lanza, M., 2018. Electronic synapses made of layered two-dimensional materials. *Nature Electronics*, 1(8), pp.458-465.
- [24] Dong, Z., Hua, Q., Xi, J., Shi, Y., Huang, T., Dai, X., Niu, J., Wang, B., Wang, Z.L. and Hu, W., 2023. Ultrafast and Low-Power 2D Bi₂O₂Se Memristors for Neuromorphic Computing Applications. *Nano Letters*, 23(9), pp.3842-3850.
- [25] Roldan, J.B., Maldonado, D., Aguilera-Pedregosa, C., Moreno, E., Aguirre, F., Romero-Zaliz, R., García-Vico, A.M., Shen, Y. and Lanza, M., 2022. Spiking neural networks based on two-dimensional materials. *npj 2D Materials and Applications*, 6(1), p.63.

- [26] Pan, C., Ji, Y., Xiao, N., Hui, F., Tang, K., Guo, Y., Xie, X., Puglisi, F.M., Larcher, L., Miranda, E. and Jiang, L., 2017. Coexistence of grain-boundaries-assisted bipolar and threshold resistive switching in multilayer hexagonal boron nitride. *Advanced functional materials*, 27(10), p.1604811.
- [27] Chen, Z., Huang, W., Zhao, W., Hou, C., Ma, C., Liu, C., Sun, H., Yin, Y. and Li, X., 2019. Ultrafast Multilevel Switching in Au/YIG/h-Si RRAM. *Advanced Electronic Materials*, 5(2), p.1800418.
- [28] Zhu, K., Pazos, S., Aguirre, F., Shen, Y., Yuan, Y., Zheng, W., Alharbi, O., Villena, M.A., Fang, B., Li, X. and Milozzi, A., 2023. Hybrid 2D–CMOS microchips for memristive applications. *Nature*, 618(7963), pp.57-62.
- [29] Wong, H.S.P., Lee, H.Y., Yu, S., Chen, Y.S., Wu, Y., Chen, P.S., Lee, B., Chen, F.T. and Tsai, M.J., 2012. Metal–oxide RRAM. *Proceedings of the IEEE*, 100(6), pp.1951-1970.
- [30] Ye, C., Wu, J., He, G., Zhang, J., Deng, T., He, P. and Wang, H., 2016. Physical mechanism and performance factors of metal oxide based resistive switching memory: a review. *Journal of Materials Science & Technology*, 32(1), pp.1-11.
- [31] Kumar, D., Aluguri, R., Chand, U. and Tseng, T.Y., 2017. Metal oxide resistive switching memory: materials, properties and switching mechanisms. *Ceramics International*, 43, pp.S547-S556.
- [32] Wu, X., Ge, R., Chen, P.A., Chou, H., Zhang, Z., Zhang, Y., Banerjee, S., Chiang, M.H., Lee, J.C. and Akinwande, D., 2019. Thinnest nonvolatile memory based on monolayer h-BN. *Advanced Materials*, 31(15), p.1806790.
- [33] Kim, M., Ge, R., Wu, X., Lan, X., Tice, J., Lee, J.C. and Akinwande, D., 2018. Zero-static power radio-frequency switches based on MoS₂ atomistors. *Nature communications*, 9(1), p.2524.
- [34] Ge, R., Wu, X., Kim, M., Shi, J., Sonde, S., Tao, L., Zhang, Y., Lee, J.C. and Akinwande, D., 2018. Atomistor: nonvolatile resistance switching in atomic sheets of transition metal dichalcogenides. *Nano letters*, 18(1), pp.434-441.
- [35] Zhao, H., Dong, Z., Tian, H., DiMarzi, D., Han, M.G., Zhang, L., Yan, X., Liu, F., Shen, L., Han, S.J. and Cronin, S., 2017. Atomically thin femtojoule memristive device. *Advanced Materials*, 29(47), p.1703232.
- [36] Huang, Y., Gu, Y., Wu, X., Ge, R., Chang, Y.F., Wang, X., Zhang, J., Akinwande, D. and Lee, J.C., 2021. ReSe₂-Based RRAM and Circuit-Level Model for Neuromorphic Computing. *Frontiers in Nanotechnology*, 3, p.782836.
- [37] Xu, R., Jang, H., Lee, M.H., Amanov, D., Cho, Y., Kim, H., Park, S., Shin, H.J. and Ham, D., 2019. Vertical MoS₂ double-layer memristor with electrochemical metallization as an atomic-scale synapse with switching thresholds approaching 100 mV. *Nano letters*, 19(4), pp.2411-2417
- [38] Shi, Y., Pan, C., Chen, V., Raghavan, N., Pey, K.L., Puglisi, F.M., Pop, E., Wong, H.S. and Lanza, M., 2017, December. Coexistence of volatile and non-volatile resistive switching in 2D h-BN based electronic synapses. In *2017 IEEE International Electron Devices Meeting (IEDM)* (pp. 5-4). IEEE.
- [39] Shi, Y., Liang, X., Yuan, B., Chen, V., Li, H., Hui, F., Yu, Z., Yuan, F., Pop, E., Wong, H.S.P. and Lanza, M., 2018. Electronic synapses made of layered two-dimensional materials. *Nature Electronics*, 1(8), pp.458-465.
- [40] Jing, X., Puglisi, F., Akinwande, D. and Lanza, M., 2019. Chemical vapor deposition of hexagonal boron nitride on metal-coated wafers and transfer-free fabrication of resistive switching devices. *2D Materials*, 6(3), p.035021.
- [41] Wang, C.H., McClellan, C., Shi, Y., Zheng, X., Chen, V., Lanza, M., Pop, E. and Wong, H.S.P., 2018, December. 3D monolithic stacked 1T1R cells using monolayer MoS₂ FET and hBN RRAM fabricated at low (150° C) temperature. In *2018 IEEE International Electron Devices Meeting (IEDM)* (pp. 22-5). IEEE.
- [42] Yuan, B., Liang, X., Zhong, L., Shi, Y., Palumbo, F., Chen, S., Hui, F., Jing, X., Villena, M.A., Jiang, L. and Lanza, M., 2020. 150 nm × 200 nm cross-point hexagonal boron nitride-based memristors. *Advanced Electronic Materials*, 6(12), p.1900115.
- [43] Yeh, C.H., Zhang, D., Cao, W. and Banerjee, K., 2020, December. 0.5 T0. 5R-Introducing an Ultra-Compact Memory Cell Enabled by Shared Graphene Edge-Contact and h-BN Insulator. In *2020 IEEE International Electron Devices Meeting (IEDM)* (pp. 12-3). IEEE.

- [44] Gu, Y., Sema, M.I., Mohan, S., Londoño-Calderon, A., Ahmed, T., Huang, Y., Lee, J., Walia, S., Pettes, M.T., Liechti, K.M. and Akinwande, D., 2022. Sulfurization Engineering of One-Step Low-Temperature MoS₂ and WS₂ Thin Films for Memristor Device Applications. *Advanced Electronic Materials*, 8(2), p.2100515.
- [45] Shen, Y., Zheng, W., Zhu, K., Xiao, Y., Wen, C., Liu, Y., Jing, X. and Lanza, M., 2021. Variability and Yield in h-BN-Based Memristive Circuits: The Role of Each Type of Defect. *Advanced Materials*, 33(41), p.2103656.
- [46] Yan, X., Zhao, Q., Chen, A.P., Zhao, J., Zhou, Z., Wang, J., Wang, H., Zhang, L., Li, X., Xiao, Z. and Wang, K., 2019. Vacancy-induced synaptic behavior in 2D WS₂ nanosheet-based memristor for low-power neuromorphic computing. *Small*, 15(24), p.1901423
- [47] Zhuang, P., Lin, W., Ahn, J., Catalano, M., Chou, H., Roy, A., Quevedo-Lopez, M., Colombo, L., Cai, W. and Banerjee, S.K., 2020. Nonpolar resistive switching of multilayer-hBN-based memories. *Advanced Electronic Materials*, 6(1), p.1900979.
- [48] Zhang, F., Zhang, H., Shrestha, P.R., Zhu, Y., Maize, K., Krylyuk, S., Shakouri, A., Campbell, J.P., Cheung, K.P., Bendersky, L.A. and Davydov, A.V., 2018, December. An ultra-fast multi-level MoTe₂-based RRAM. In *2018 IEEE International Electron Devices Meeting (IEDM)* (pp. 22-7). IEEE
- [49] Tang, B., Veluri, H., Li, Y., Yu, Z.G., Waqar, M., Leong, J.F., Sivan, M., Zamburg, E., Zhang, Y.W., Wang, J. and Thean, A.V., 2022. Wafer-scale solution-processed 2D material analog resistive memory array for memory-based computing. *Nature Communications*, 13(1), p.3037.
- [50] Wang, M., Cai, S., Pan, C., Wang, C., Lian, X., Zhuo, Y., Xu, K., Cao, T., Pan, X., Wang, B. and Liang, S.J., 2018. Robust memristors based on layered two-dimensional materials. *Nature Electronics*, 1(2), pp.130-136.
- [51] Braun, D., Lukas, S., Völkel, L., Hartwig, O., Prechtel, M., Belete, M., Kataria, S., Wahlbrink, T., Daus, A., Duesberg, G.S. and Lemme, M.C., 2022, June. Non-Volatile Resistive Switching in PtSe₂-Based Crosspoint Memristors. In *2022 Device Research Conference (DRC)* (pp. 1-2). IEEE.
- [52] Jeong, H.Y., Kim, J.Y., Kim, J.W., Hwang, J.O., Kim, J.E., Lee, J.Y., Yoon, T.H., Cho, B.J., Kim, S.O., Ruoff, R.S. and Choi, S.Y., 2010. Graphene oxide thin films for flexible nonvolatile memory applications. *Nano letters*, 10(11), pp.4381-4386.
- [53] Wang, Y., Yang, J., Wang, Z., Chen, J., Yang, Q., Lv, Z., Zhou, Y., Zhai, Y., Li, Z. and Han, S.T., 2019. Near-infrared annihilation of conductive filaments in quasiplane MoSe₂/Bi₂Se₃ nanosheets for mimicking heterosynaptic plasticity. *Small*, 15(7), p.1805431

Reviewer 3

It would be useful if the authors could elaborate on the proposed switching mechanism. The implication that a Ti filament is responsible for switching (lines 99-103) a very interesting point, but only briefly mentioned. If cationic motion of Ti is responsible for the filament, perhaps this is a differentiating feature of 2D-based memristors from the traditional TMO memristors which are (typically) thought to be modulated by oxygen vacancy anions. I wonder if the physical insight here is applicable to traditional TMO memristors.

Thank you for the comment. The switching mechanism of our devices is based on the formation and dissolution of titanium ion-based filaments, as suggested by EELS profile (Fig.2g in main text) and previous studies [1-2]. During the SET process, the applied electric field lowers the energy barrier required to release Ti ions from the electrode interface. These released Ti ions migrate into the hBN switching layer (aided by the electric field) and form conductive filaments connecting both electrodes. The current passing through the device generates temperature in the filament through joule heating. This temperature lowers the energy barrier required for ion release, further propelling the filament growth. During the RESET process, the joule heating and the electric field dissolve the filament, and the ions are driven back to the electrode. As suggested, we have included additional text to clearly explain the switching mechanism (lines 169-173 main text).

We would like to mention that this similar metal ion based switching mechanism has been reported even using traditional TMO memristors commonly referred to as “Conductive bridge RAMs (CBRAM)”. Although the original CBRAMs were proposed using chalcogenide-based (Cu-GeSe_x, Ag-GeSe_x) solid electrolytes [3-6], subsequently, CBRAMs using traditional oxides such as SiO₂, HfO₂, Al₂O₃, Ta₂O₅, TiO₂ have been explored [7-15]. Therefore, although the switching mechanism in our devices is different from oxygen vacancy based OxRAMs, but it is very similar to the CBRAMs.

Supplementary Table 1 is a useful reference comparing switching in standard TMO and 2D memristor. I am wondering if, based on this work, the authors can further comment on fundamental differences between switching in traditional versus 2D materials. I realize there is quite a bit of interest in the novelty of 2D materials, but am wondering if there are fundamental advantages of the 2D stack. My impression from this work and refs in the Table is that that 2D memristors have similar switching speed and energy, but the endurance may be somewhat worse.

Thank you for the valuable and insightful comment. To answer this question, we conducted transient thermal studies using a finite element method based physics solver (COMSOL). The results show that, indeed, there are fundamental advantages to using 2D memristors from a switching speed perspective. On the endurance front, we agree that the state-of-the-art 2D memristors fail to exhibit on-par performance with TMO memristors. However, 2D memristors are in the nascent stages of development, with many challenges associated with CVD growth, transfer process, and metal electrode roughness still being explored. The following paragraphs delve into the discussion on switching speed and endurance in greater detail.

Switching speed

Quantitatively, the results presented in this work, i.e., 600 cycles with 1.43ns switching speed and 100 cycles with 120ps switching speed, may not seem like a big improvement from previous

reports (20 cycles at 120ps). However, this work is very significant as it conclusively proves the ability of 2D memristors to switch consistently with ultra-fast voltage pulses. It should be noted that testing in a sub-nanosecond regime is extremely difficult (test setup and analysis) and complicated. Therefore, most studies typically characterize the switching speed based on a single experiment. Characterizing the memristor speed based on a single cycle is unreliable and potentially misleading. It is difficult to ascertain that the observed fast response is associated with a reversible switching process. For example, a device shorted (failed) by the fast pulse will produce a current signature similar to SET. Likewise, a device whose current leads are damaged due to excessive current (electromigration) will produce a signature similar to RESET. Therefore, the switching speed of the device should only be characterized with considering subsequent cycling data. In this regard, only two research articles show repeatable switching with 10 cycles [15] and 20 cycles [16] using ultra-short voltage pulses. The fact that many studies [17-27] do not show cycling data may suggest that those devices do not switch consistently. On the other hand, we could present statistical data which shows that ultra-fast switching in 2D memristors is intrinsic and not just a rare occurrence.

The switching mechanism in our devices is based on metal ion diffusion into the switching layer. The applied electric field facilitates the vertical propagation of the filament around weak spots in the dielectric. This initially formed narrow filament connects the electrodes and lowers the resistance of the memristor. The current flowing through this filament generates temperature through the Joule heating phenomenon. The temperature generated in the filament lowers the energy barrier for ion release from the electrode and further propels the filament growth. To understand the fundamental reason for the ultra-fast switching in our devices, we perform transient thermal analysis using finite element based physics solvers (COMSOL). To summarize the results, the characteristic properties aiding the ultra-fast switching in 2D hBN memristors are:

1. Ultra-thin switching layer
2. High thermal conductivity of 2D hBN layers

Ultra-thin switching layer

Oxide memristors (commercial or university demonstrations) typically have a thicker switching layer (5nm-20nm) in comparison to the memristors in this study (< 2.4nm). The thinner switching layer significantly increases the local electric field across the dielectric, which promotes faster resistance switching. Fig.R3.1a shows the electric field increase inside the switching layer as the dielectric thickness reduces.

Figure R3.1 (a) Electric field vs dielectric thickness for $V=1V$. (b) Snapshot of the COMSOL model showing the metallic filament and electrodes. The temperature profile shows the highest temperature at the center of the filament. (c) Temperature reaching the interface vs dielectric thickness (d) time taken for the interface to reach 400K, 500K, 600K as a function of dielectric thickness.

From the Arrhenius model (Eq.R2.1,2), the filament growth dynamics depend on the temperature in addition to the applied voltage. The critical physical process that determines the switching speed of the device is the metal ion release [28] which is determined by the Ti/hBN interface temperature (T_{INT}). Note that this temperature is different from the filament temperature (T_{CF}). The current passing through the device generates temperature in the filament (T_{CF}) through joule heating. This temperature is highest at the center of the filament (vertically) and lowers towards the electrodes, as seen in Fig.R3.1b. This nature of the temperature profile has been reported in several previous studies [29-31]. Most studies approximate $T_{CF} \sim T_{INT}$ for simplicity in analytical modeling.

Using the COMSOL model, we study the T_{INT} dependence on dielectric thickness, as shown in Fig.R3.1c. It is evident that as the dielectric thickness reduces, steady-state T_{INT} increases. In addition, Fig.R3.1d plots the time taken for temperature build-up at the interface as a function of dielectric thickness. We plot the time taken to reach 400K, 500K, and 600K for comparison. The T_{INT} for thinner dielectric (such as the devices in this study) reaches 600K in less than 1ps, whereas thicker dielectric ~ 5 nm takes about 40ps. The faster temperature build-up at the interface promotes faster metal ion release, which creates a positive feedback loop: higher $T_{INT} \rightarrow$ lower $R \rightarrow$ higher T_{INT} . This positive feedback loop promotes faster switching in devices with a thinner dielectric layer.

High thermal conductivity of 2D hBN

Based on the previous discussion, the obvious subsequent question would be - "Will the oxide RRAMs with thinner switching layer switch as fast as the 2D memristors?" To answer this question, we simulated two devices – one with an HfO_x layer and the other with 2D hBN, with identical switching layer thicknesses (2.5nm).

Figure R3.2 (a) Snapshot of COMSOL model showing three points of interest A,B,C. Transient interface temperature at (b) point A (c) point B (d) point C.

In addition to thickness, another physical property that determines the switching speed of the devices is the thermal conductivity of the switching layer. The in-plane thermal conductivity of hBN ($k \sim 100\text{-}200\text{ W/mK}$) [32-34] is significantly higher than oxides ($k \sim 0.5\text{-}2\text{ W/mK}$) typically used for memristors (HfO_2 , Al_2O_3 , TaO_x) [35-37]. On the other hand, the out-of-plane thermal conductivity is comparable ($5\text{-}10\text{ mW/K}$) to the oxides. The high thermal conductivity quickly spreads the heat from the filament to the surroundings, which heats a larger cross-section of the interface. Fig.R3.2a shows three points on the interface at 2nm,4nm, and 6nm away from the interface. Temperature transients at these points (Fig.R3.2b,c,d) highlight this effect. Evidently, T_{INT} increases rapidly for hBN compared to HfO_x . This effect is especially more pronounced as we move away from the filament (points B, C). Since a larger cross-section of the interface is heated, the probability of ion release increases, which results in faster filament expansion. Therefore, 2D materials that typically have higher thermal conductivity promote faster switching compared to oxide RRAMs with identical layer thickness.

Overall, the 2D memristors benefit from having an ultra-thin switching layer as well as high thermal conductivity compared to TMO memristors. Fig.R3.3 shows the T_{INT} for hBN devices with 2.5nm thickness and HfO_x devices with 5nm thickness (typical thickness for TMO memristors). Clearly, the temperature rises rapidly for the hBN devices resulting in ultra-fast switching.

This explanation for ultra-fast switching observed in 2D memristors has been added as Supplementary Figure.16 in the manuscript.

Figure R3.3 Average transient interface temperature for hBN with thickness=2.5nm and HfOx with thickness=5nm

Endurance

The endurance in 2D memristors has been a concern, mainly attributed to the ultra-thin switching layer. Transition metal oxide (TMO) based memristors typically have a switching layer thickness ranging between 5nm-20nm [38-40]. On the contrary, 2D memristors have thinner switching layers. The reduced vertical separation between metal electrodes enhances the electric field, which results in reduced control over the metal ion migration into the switching layer and subsequent filament formation. Moreover, as the thickness of the switching layer reduces, the impact of bottom/top electrode roughness plays a significant role in determining the overall device performance.

A very recent study [41] reported high endurance of $\sim 10^6$ cycles in the 1T1M (1transistor-1memristor) bitcell with ~ 6.6 nm thick switching layer where the transistor acts as a current compliance device limiting excess ion diffusion into the memristor. In that study, the standalone devices without CMOS transistors exhibited only 100 cycles, whereas devices with transistors exhibited a million cycles. This promising result suggests that the adjoining selector device can improve endurance in 2D memristors. The characterization setup utilized in this study cannot support external current compliance (CC) for high-frequency pulse testing. For the frequency range studied in this article, the external CC device (resistor, diode or transistor) must be located within <1 cm distance from the DUT (device under test) to be considered as a lumped circuit element. Therefore, the only possible option is to have a CC device fabricated monolithically, like in [41]. Without current compliance, most of our devices get shorted during testing, which cannot be recovered. We expect that having a 1T1M configuration would improve the endurance of 2D memristors.

Overall, 2D memristors show tremendous potential to outperform traditional memristors in all aspects – operating voltage, switching speed, retention, and endurance. In addition, as we know, 2D memristors offer other advantages such as transparency, flexibility, and mechanical strength. We firmly believe that as advances are made in 2D technology, such as reliable large-area uniform CVD growth, contaminant-free transfer process etc., superior memristors will be realized.

The resistance evolution in Fig 4d does not appear to align with the level described in the text. On lines 154, the initial range is 10k-100kohms, but in Fig 4d, we see the evolution of resistance in a much lower range, from about 700-1000 to 200 ohms. If the authors have data showing SET switching starting from the higher range, that would be of interest.

Thank you for the comment.

This discrepancy in the resistance range arises from the intricacies of the test setup. Most commercial high-frequency oscilloscopes (>10GHz) unfortunately only support an input impedance of 50Ω. With an input impedance of 50Ω, currents below 0.2mA translate to voltages below 10mV, which appear as noise on the scope. Due to this limitation, we have included a current amplifier to read the high resistances of the memristors. However, this current amplifier can only operate at lower frequencies (<5MHz), because of which we place it on the DC path. So, we can read high resistance levels (>10⁴ Ω) before/after the pulse, but the dynamic evolution of resistance is only perceivable when the resistance drops to lower values (< 1000 Ω). Therefore, we model the resistance evolution post-pulse rise time. We have mentioned this in the text and the figure caption in the manuscript. As rightly pointed out, it would be interesting if we could capture and study the resistance evolution across the entire range of resistance change.

Minor: In general the variables are carefully defined for the model in the supplement, but I would recommend adding definitions for supplementary Eqn 6.

Thank you for pointing this out. One of the other reviewers suggested we update the section on the resistive switching model. So, the sub-section focusing on the thermal time constant (which includes Eq.6) has been removed from the manuscript.

Minor: The plot text on several figures is too small and hard to read (esp insets).

Thank you for suggestions that help improve the readability of the manuscript. We have increased the plot text on all the figures and associated insets.

References

- [1] Shi, Y., Liang, X., Yuan, B., Chen, V., Li, H., Hui, F., Yu, Z., Yuan, F., Pop, E., Wong, H.S.P. and Lanza, M., 2018. Electronic synapses made of layered two-dimensional materials. *Nature Electronics*, 1(8), pp.458-465.
- [2] Puglisi, F.M., Larcher, L., Pan, C., Xiao, N., Shi, Y., Hui, F. and Lanza, M., 2016, December. 2D h-BN based RRAM devices. In *2016 IEEE International Electron Devices Meeting (IEDM)* (pp. 34-8). IEEE.
- [3] Kozicki, M.N., Park, M. and Mitkova, M., 2005. Nanoscale memory elements based on solid-state electrolytes. *IEEE Transactions on Nanotechnology*, 4(3), pp.331-338.
- [4] Kozicki, M.N., Gopalan, C., Balakrishnan, M., Park, M. and Mitkova, M., 2004, November. Nonvolatile memory based on solid electrolytes. In *Proceedings. 2004 IEEE Computational Systems Bioinformatics Conference* (pp. 10-17). IEEE.
- [5] Soni, R., Meuffels, P., Staikov, G., Weng, R., Kügeler, C., Petraru, A., Hambe, M., Waser, R. and Kohlstedt, H., 2011. On the stochastic nature of resistive switching in Cu doped Ge_{0.3}Se_{0.7} based memory devices. *Journal of applied physics*, 110(5).
- [6] Rahaman, S.Z., Maikap, S., Chiu, H.C., Lin, C.H., Wu, T.Y., Chen, Y.S., Tzeng, P.J., Chen, F., Kao, M.J. and Tsai, M.J., 2010. Bipolar resistive switching memory using Cu metallic filament in Ge_{0.4}Se_{0.6} solid electrolyte. *Electrochemical and Solid-State Letters*, 13(5), p.H159.

- [7] Thermadam, S.P., Bhagat, S.K., Alford, T.L., Sakaguchi, Y., Kozicki, M.N. and Mitkova, M., 2010. Influence of Cu diffusion conditions on the switching of Cu–SiO₂-based resistive memory devices. *Thin Solid Films*, 518(12), pp.3293-3298.
- [8] Tappertzhofen, S., Mündelein, H., Valov, I. and Waser, R., 2012. Nanoionic transport and electrochemical reactions in resistively switching silicon dioxide. *Nanoscale*, 4(10), pp.3040-3043.
- [9] Goux, L., Sankaran, K., Kar, G., Jossart, N., Opsomer, K., Degraeve, R., Pourtois, G., Rignanese, G.M., Detavernier, C., Clima, S. and Chen, Y.Y., 2012, June. Field-driven ultrafast sub-ns programming in WAl₂O₃/Ti/CuTe-based 1T1R CBRAM system. In *2012 Symposium on VLSI Technology (VLSIT)* (pp. 69-70). IEEE.
- [10] Belmonte A 2015 Electrical characterization and physical modeling of novel CBRAM stacks PhD Dissertation Katholieke Universiteit Leuven
- [11] Tsuruoka, T., Terabe, K., Hasegawa, T. and Aono, M., 2011. Temperature effects on the switching kinetics of a Cu–Ta₂O₅-based atomic switch. *Nanotechnology*, 22(25), p.254013.
- [12] Tsuruoka, T., Hasegawa, T., Valov, I., Waser, R. and Aono, M., 2013. Rate-limiting processes in the fast SET operation of a gapless-type Cu-Ta₂O₅ atomic switch. *AIP Advances*, 3(3).
- [13] Tsunoda, K., Fukuzumi, Y., Jameson, J.R., Wang, Z., Griffin, P.B. and Nishi, Y., 2007. Bipolar resistive switching in polycrystalline TiO₂ films. *Applied physics letters*, 90(11).
- [14] Haemori, M., Nagata, T. and Chikyow, T., 2009. Impact of Cu electrode on switching behavior in a Cu/HfO₂/Pt structure and resultant Cu ion diffusion. *Applied Physics Express*, 2(6), p.061401.
- [15] Choi, B.J., Torrezan, A.C., Norris, K.J., Miao, F., Strachan, J.P., Zhang, M.X., Ohlberg, D.A., Kobayashi, N.P., Yang, J.J. and Williams, R.S., 2013. Electrical performance and scalability of Pt dispersed SiO₂ nanometallic resistance switch. *Nano letters*, 13(7), pp.3213-3217.
- [16] Chen, Z., Huang, W., Zhao, W., Hou, C., Ma, C., Liu, C., Sun, H., Yin, Y. and Li, X., 2019. Ultrafast Multilevel Switching in Au/YIG/n-Si RRAM. *Advanced Electronic Materials*, 5(2), p.1800418
- [17] Wang, C., Wu, H., Gao, B., Wu, W., Dai, L., Li, X. and Qian, H., 2017. Ultrafast RESET analysis of HfO_x-based RRAM by sub-nanosecond pulses. *Advanced Electronic Materials*, 3(12), p.1700263.
- [18] Shrestha, P.R., Nminibapiel, D., Kim, J.H., Campbell, J.P., Cheung, K.P., Deora, S., Bersuker, G. and Baumgart, H., 2014, June. Energy control paradigm for compliance-free reliable operation of RRAM. In *2014 IEEE International Reliability Physics Symposium* (pp. MY-10). IEEE.
- [19] Choi, B.J., Torrezan, A.C., Strachan, J.P., Kotula, P.G., Lohn, A.J., Marinella, M.J., Li, Z., Williams, R.S. and Yang, J.J., 2016. High-speed and low-energy nitride memristors. *Advanced Functional Materials*, 26(29), pp.5290-5296.
- [20] Lee, H.Y., Chen, Y.S., Chen, P.S., Gu, P.Y., Hsu, Y.Y., Wang, S.M., Liu, W.H., Tsai, C.H., Sheu, S.S., Chiang, P.C. and Lin, W.P., 2010, December. Evidence and solution of over-RESET problem for HfO_x based resistive memory with sub-ns switching speed and high endurance. In *2010 International Electron Devices Meeting* (pp. 19-7). IEEE.
- [21] Torrezan, A.C., Strachan, J.P., Medeiros-Ribeiro, G. and Williams, R.S., 2011. Sub-nanosecond switching of a tantalum oxide memristor. *Nanotechnology*, 22(48), p.485203.
- [22] Böttger, U., von Witzleben, M., Havel, V., Fleck, K., Rana, V., Waser, R. and Menzel, S., 2020. Picosecond multilevel resistive switching in tantalum oxide thin films. *Scientific reports*, 10(1), pp.1-9.
- [23] Zhang, F., Zhang, H., Shrestha, P.R., Zhu, Y., Maize, K., Krylyuk, S., Shakouri, A., Campbell, J.P., Cheung, K.P., Bendersky, L.A. and Davydov, A.V., 2018, December. An ultra-fast multi-level MoTe₂-based RRAM. In *2018 IEEE International Electron Devices Meeting (IEDM)* (pp. 22-7). IEEE.
- [24] Ge, R., Wu, X., Kim, M., Chen, P.A., Shi, J., Choi, J., Li, X., Zhang, Y., Chiang, M.H., Lee, J.C. and Akinwande, D., 2018, December. Atomrators: Memory effect in atomically-thin sheets and record RF switches. In *2018 IEEE International Electron Devices Meeting (IEDM)* (pp. 22-6). IEEE.
- [25] Kim, M., Ducournau, G., Skrzypczak, S., Yang, S.J., Szriftgiser, P., Wainstein, N., Stern, K., Happy, H., Yalon, E., Pallecchi, E. and Akinwande, D., 2022. Monolayer molybdenum disulfide switches for 6G communication systems. *Nature Electronics*, pp.1-7.
- [26] Wu, X., Ge, R., Kim, M., Akinwande, D. and Lee, J.C., 2020, February. Atomrators: Non-volatile resistance switching in 2D monolayers. In *2020 Pan Pacific Microelectronics Symposium (Pan Pacific)* (pp. 1-6). IEEE.

- [27] Zhang, F., Zhang, H., Krylyuk, S., Milligan, C.A., Zhu, Y., Zemlyanov, D.Y., Bendersky, L.A., Burton, B.P., Davydov, A.V. and Appenzeller, J., 2019. Electric-field induced structural transition in vertical MoTe₂-and Mo_{1-x}W_xTe₂-based resistive memories. *Nature materials*, 18(1), pp.55-61.
- [28] Huang, P., Liu, X.Y., Chen, B., Li, H.T., Wang, Y.J., Deng, Y.X., Wei, K.L., Zeng, L., Gao, B., Du, G. and Zhang, X., 2013. A physics-based compact model of metal-oxide-based RRAM DC and AC operations. *IEEE transactions on electron devices*, 60(12), pp.4090-4097.
- [29] Niraula, D. and Karpov, V.G., 2017. Heat transfer in filamentary RRAM devices. *IEEE Transactions on Electron Devices*, 64(10), pp.4106-4113.
- [30] Ielmini, D., Nardi, F. and Cagli, C., 2011. Physical models of size-dependent nanofilament formation and rupture in NiO resistive switching memories. *Nanotechnology*, 22(25), p.254022.
- [31] Larentis, S., Nardi, F., Balatti, S., Gilmer, D.C. and Ielmini, D., 2012. Resistive switching by voltage-driven ion migration in bipolar RRAM—Part II: Modeling. *IEEE Transactions on Electron Devices*, 59(9), pp.2468-2475.
- [32] Jo, I., Pettes, M.T., Kim, J., Watanabe, K., Taniguchi, T., Yao, Z. and Shi, L., 2013. Thermal conductivity and phonon transport in suspended few-layer hexagonal boron nitride. *Nano letters*, 13(2), pp.550-554.
- [33] Alam, M.T., Bresnehan, M.S., Robinson, J.A. and Haque, M.A., 2014. Thermal conductivity of ultra-thin chemical vapor deposited hexagonal boron nitride films. *Applied Physics Letters*, 104(1).
- [34] Jana, M. and Singh, R.N., 2018. Progress in CVD synthesis of layered hexagonal boron nitride with tunable properties and their applications. *International Materials Reviews*, 63(3), pp.162-203.
- [35] Panzer, M.A., Shandalov, M., Rowlette, J.A., Oshima, Y., Chen, Y.W., McIntyre, P.C. and Goodson, K.E., 2009. Thermal properties of ultrathin hafnium oxide gate dielectric films. *IEEE Electron Device Letters*, 30(12), pp.1269-1271.
- [36] Cappella, A., Battaglia, J.L., Schick, V., Kusiak, A., Lamperti, A., Wiemer, C. and Hay, B., 2013. High Temperature Thermal Conductivity of Amorphous Al₂O₃ Thin Films Grown by Low Temperature ALD. *Advanced Engineering Materials*, 15(11), pp.1046-1050.
- [37] Landon, C.D., Wilke, R.H., Brumbach, M.T., Brennecke, G.L., Blea-Kirby, M., Ihlefeld, J.F., Marinella, M.J. and Beechem, T.E., 2015. Thermal transport in tantalum oxide films for memristive applications. *Applied Physics Letters*, 107(2).
- [38] Wong, H.S.P., Lee, H.Y., Yu, S., Chen, Y.S., Wu, Y., Chen, P.S., Lee, B., Chen, F.T. and Tsai, M.J., 2012. Metal-oxide RRAM. *Proceedings of the IEEE*, 100(6), pp.1951-1970.
- [39] Ye, C., Wu, J., He, G., Zhang, J., Deng, T., He, P. and Wang, H., 2016. Physical mechanism and performance factors of metal oxide based resistive switching memory: a review. *Journal of Materials Science & Technology*, 32(1), pp.1-11.
- [40] Kumar, D., Aluguri, R., Chand, U. and Tseng, T.Y., 2017. Metal oxide resistive switching memory: materials, properties and switching mechanisms. *Ceramics International*, 43, pp.S547-S556
- [41] Zhu, K., Pazos, S., Aguirre, F., Shen, Y., Yuan, Y., Zheng, W., Alharbi, O., Villena, M.A., Fang, B., Li, X. and Milozzi, A., 2023. Hybrid 2D-CMOS microchips for memristive applications. *Nature*, 618(7963), pp.57-62.

Reviewer 4

Theory: In my opinion the theoretical model is a valuable work but cannot be accepted as a close model of what happens in experiments. For example, 3 nm filament has very similar thickness to hBN layer. The materials of filament is not clear. There might be a population of filaments at the interface to electrode with different sizes. There is a complicated heat bath (generation and dissipation) that is not argued. What are authors looking for and what consequences are expected with such a model is not clear. Way VCM and TCM are not modeled and only attention is made on filament formation.

Thank you for the valuable and insightful comment.

The switching mechanism in 2D memristors was reported to be based on the formation and dissolution of nano-conductive filaments, as observed through TEM imaging [1]. In our experiments, the EELS profile (Fig.2g main text) revealed the presence of Ti ions within the hBN layer, suggesting Ti ion constituted nano-conductive filaments. Similar observations of Ti ion diffusion into the hBN layer were reported in previous studies [1-2]. Moreover, the 2D films synthesized using CVD growth inherently possess grain boundaries and defects (main text Fig.1e), that serve as favorable sites for conductive filaments. Notably, few studies report the absence of resistive switching behavior in memristors fabricated using exfoliated flakes that lack intrinsic defects [3-4]. If the switching mechanism were based on VCM or TCM, resistive switching behavior should have been observed in these exfoliated flake memristors. Taking these observations into account, along with our own experimental findings, we believe that the switching mechanism in our devices is based on active metal ion migration into the switching layer, similar to conductive-bridge RAM (CBRAM) [5-6]. Therefore, in this study, we do not consider other switching mechanisms, such as VCM or TCM.

Irrespective of the filament composition, the growth dynamics can be modeled using a universal phenomenological based on the Arrhenius relationship [7-10]. This model is widely accepted in the field of memristors and has been employed to study TMO and CBRAM memristors [7-10]. Therefore, in this study, we employ the same model with a few modifications to capture the growth dynamics as well as the Joule heating induced filament narrowing (Fig.5c main text) in our 2D memristors.

The switching mechanism in our devices can be summarized as follows: the applied electric field lowers the energy barrier required to release Ti ions from the electrode interface. These released Ti ions migrate into the hBN switching layer (aided by the electric field) and form conductive filaments connecting both electrodes. As rightly pointed out, there is a complicated heat bath inside the memristor that supports different physical phenomena. Previously reported models [7-10] consider filament formation and dissolution as independent events i.e., during a SET process, all the equations only model filament expansion. However, in reality, filament formation and dissolution proceed simultaneously (determined by voltage and temperature), albeit at different rates. For example, as the filament begins to widen, the temperature generated in the filament increases, which promotes outward diffusion of the metal ions from the filament. Since the memristors in this study have an ultra-thin switching layer (<2.5nm), very low resistance filaments (200Ω -500Ω) are formed during the SET process. These low-resistance filaments generate high filament temperatures that promote dissolution resulting in filament narrowing. This phenomenon of filament narrowing was evidently observed in our devices (Fig.R4.1). For this study, we measured the resistance of the device before (R_{INIT}), after (R_{FIN}), and during the pulse (R_{PUL}). The

resistance distributions of R_{INIT} , R_{PUL} , and R_{FIN} (Fig.R4.1b) show that the memristor resistance reduces during the SET pulse and again increases after the pulse suggesting a reduction in filament cross-section.

The Arrhenius models (Eq.R4.1) [7-10] can only capture filament expansion during the SET process and vice versa during the RESET process. Therefore, the final resistance estimated by these models will need to be revised.

$$\frac{d\Phi}{dt} = Ae^{\left(-\frac{Ea_0 - \alpha qV}{kT_{CF}}\right)} \quad \text{----- (R4.1)}$$

To account for filament narrowing, we included a second term that captures the dynamics of filament dissolution (Eq.R4.2).

$$\frac{d\Phi}{dt} = A_1e^{\left(-\frac{Ea_0 - \alpha qV}{kT_{CF}}\right)} - A_2e^{\left(-\frac{Ea}{kT_{CF}}\right)} \quad \text{----- (R4.2)}$$

Figure R4.1 (a) Voltage and current pulse waveforms during a SET operation. (b) Resistance distribution of R_{INIT} , R_{PUL} , R_{FIN} . The R_{PUL} was estimated as the resistance value calculated at the maximum current magnitude during the pulse. The resistance value is calculated at the maximum current magnitude during the pulse (R_{PUL}) and immediately before (R_{INIT}) and after (R_{FIN}) the pulse via applying a small DC bias $V=100mV$.

Note that the activation energy Ea required for metal ion out-diffusion from the filament differs from activation energy for filament formation (bond breaking and ion hopping). As the voltage pulse ramps up, the first term dominates, and the filament expands. After reaching a certain filament width, the filament growth rate is balanced by the filament dissolution rate, and the filament growth saturates. However, during the voltage pulse falling transition, the growth rate (determined by V , T_{CF}) recedes faster than the dissolution rate (determined only by T_{CF}), thereby resulting in the narrowing of the filament.

Figure R4.2 (a) Voltage and current pulse waveforms during a SET operation. The model current closely traces the measured current. Calculated time evolution of (b) filament diameter (c) memristor resistance.

A self-consistent electro-thermal solver using the proposed model clearly captures the filament narrowing (Fig.R4.2b,c). The filament expands in diameter with the pulse and narrows after pulse termination. The memristor resistance change closely matches the resistance values presented in Fig.R4.1b. We believe that this joule heating driven filament narrowing will be observed in other 2D memristors as well, and therefore the proposed equations can be used to model them. We would like to point out that, this is one of the very few studies that considers generation and dissipation processes taking place simultaneously.

Similarly, the filament dynamics during RESET can be captured using Eq.R4.3,

$$\frac{d\phi}{dt} = -A_1 e^{\left(-\frac{Ea_0 - \alpha qV}{kT_{CF}}\right)} - A_2 e^{\left(-\frac{Ea}{kT_{CF}}\right)} \quad \text{----- (R4.3)}$$

Here, the term associated with Joule heating accelerates filament dissolution and aids the RESET process. The electro-thermal solver using the above equation closely models the measured RESET current signature as seen from Fig.R4.3a. We have added additional text in the supplementary and main text to emphasize the importance of the proposed model in effectively capturing filament dynamics in 2D memristors.

Figure R4.3 (a) Voltage and current pulse waveforms during a RESET operation. The model current closely traces the measured current. Calculated time evolution of (b) filament diameter (c) memristor resistance.

Experiment: There is no motivation to accept the reported record of fast switching in this layer as there is not a big change w.r.t. the previous studies. The finding of paper is not clear. What is new? statics? fast switching? thermal and heat model?

Thank you for the comment.

Quantitatively, the results presented in this work, i.e., 600 cycles with 1.43ns switching speed and 100 cycles with 120ps switching speed, may not seem like a big improvement from previous reports (20 cycles at 120ps). However, this work is very significant as it conclusively proves the ability of 2D memristors to switch consistently with ultra-fast voltage pulses. It should be noted that testing in a sub-nanosecond regime is extremely difficult (test setup and analysis) and complicated. Therefore, most studies typically characterize the switching speed based on a single experiment. Characterizing the memristor speed based on a single cycle is unreliable and potentially misleading. It is difficult to ascertain that the observed fast response is associated with a reversible switching process. For example, a device shorted (failed) by the fast pulse will produce a current signature similar to SET. Likewise, a device whose current leads are damaged due to excessive current (electromigration) will produce a signature similar to RESET. Therefore, the switching speed of the device should only be characterized by considering subsequent cycling data. In this regard, only two research articles show repeatable switching with 10 cycles [11] and 20 cycles [12] using ultra-short voltage pulses. The fact that many studies [13-20] do not show cycling data may suggest that those devices do not switch consistently.

The major contributions of this work are,

1. The devices presented in this work switch consistently with sub-nanosecond pulses. This shows the potential of these devices to be used for high-frequency applications.
2. The test setup and experimental methodology employed in this work allow us to study the dynamic evolution of resistance (during the pulse) as well as the change in resistance (before and after the pulse). To the best of our knowledge, this is the first instance where such an approach has been adopted. Correlating these data points provided some key insights into the switching mechanism, which would otherwise remain undetected. For example, a significant presence of joule heating induced filament narrowing was observed in our experiments.
3. The statistical analysis of transient switching characteristics presented in this work is novel and has never been attempted. Statistical data unveiled some unique trends such as switching time-resistance ratio, switching energy-resistance ratio etc.,.
4. We present an electro-thermal model to capture the switching characteristics as well as the secondary effects (joule heating). In addition, the fundamental reason for the ultra-fast switching nature of the 2D memristors has been explored with the help of finite element based physics solver.

Overall, we attempt to holistically study the ultra-fast switching characteristics of the 2D memristors through experiments, modeling and statistical analysis. We have updated the text in the Introduction section to highlight the primary findings of the paper.

Technical points: intro: There is repeated theme of fast electronics or switching application is different paragraphs and there is no focus. There is not a coherent introduction that what are authors plan to present in the paper. experiment: i) a high resolution image of filament formation (for different biases) is lacking, ii) in the calculation of power, since current and thus impedance changes, how authors are sure to have a delivery of power to/from different resistance loads? Could part of power be reflected? What are high risk points that might not be considered?

Thank you for the comment that helps us to improve the quality of the manuscript. As suggested, we have updated the Introduction section to be more coherent and to clearly highlight the contribution of our work.

(i) TEM image

Figure R4.4 TEM images showing single-atom defects (arrows) and few atom wide amorphous regions (dashed circles).

Obtaining TEM image of the filament through HRTEM is challenging as the probability that the FIB cut exactly passes through the filament is completely random. Unfortunately, even after extensive searching across the FIB cuts, we did not observe filaments in our experiments. Moreover, all the elements involved in our devices (B,N,Ti) have relatively smaller atomic weights that do not produce discernible differences in contrast. Although some discontinuities in the layered hBN film are observed, these can, at best, be characterized as some defects or amorphous regions rather than as conductive filaments. Even if a filament existed in the FIB cut (typical depth $\sim 100\text{nm}$), it may not be visible in TEM image unless its dimension is comparable to the depth. Since the 2D memristors have ultra-thin switching layer, narrow filaments can carry large currents compared to the thicker TMO memristors. Therefore, such large filaments typically are not created in 2D memristors (typical filament dimension $\sim 10\text{-}20\text{nm}$ [21]). The low contrast of Ti combined with the small dimension make filament imaging in 2D memristors very challenging. All the previous studies (2D materials or otherwise) that have successfully imaged filaments through HRTEM have Ag, Au, or some other heavy elements that can produce sufficient contrast [21-24]. Other noted studies [25-26] in the field using hBN and Ti material system report just defects/amorphous regions (similar to Fig.R4.4) rather than filaments. Which proves that filament imaging in Ti/hBN system is difficult.

(ii) Reflected power

Pulse characterization of the memristors was conducted in a custom in-house RF measurement setup. The nano-second pulses were generated using a Picosecond Pulse Labs 10070A pulse generator. These ultra-short pulses were delivered to the memristors via an impedance (50Ω) matched network to eliminate the reflections. A 50Ω termination resistor was soldered onto the probe-tip (Fig.R4.5) which minimized reflections and ensured maximum power transfer to the device.

Figure R4.5 Test setup schematic showing the 50Ω termination placed near the probe tips.

The pulse delivered to the device was monitored by sampling the input transmission line through a pick-off tee. The 50Ω termination resistor effectively suppresses reflections, as seen in Fig.R4.6a,c for SET and REST pulses, respectively. Whereas the reflections are significant when the 50Ω resistor was not soldered onto the probe tips, as seen in Fig.R4.6b,d. All the experiments presented in this work were conducted with the 50Ω termination on the probe tips ensuring minimal reflected power.

Figure R4.6 Transient waveform for (a) SET pulse with 50Ω resistor (b) SET pulse without 50Ω resistor (c) RESET pulse with 50Ω resistor (d) RESET pulse without 50Ω resistor.

References

1. Shi, Y., Liang, X., Yuan, B., Chen, V., Li, H., Hui, F., Yu, Z., Yuan, F., Pop, E., Wong, H.S.P. and Lanza, M., 2018. Electronic synapses made of layered two-dimensional materials. *Nature Electronics*, 1(8), pp.458-465
2. Puglisi, F.M., Larcher, L., Pan, C., Xiao, N., Shi, Y., Hui, F. and Lanza, M., 2016, December. 2D h-BN based RRAM devices. In *2016 IEEE International Electron Devices Meeting (IEDM)* (pp. 34-8). IEEE.
3. Shen, Y., Zheng, W., Zhu, K., Xiao, Y., Wen, C., Liu, Y., Jing, X. and Lanza, M., 2021. Variability and Yield in h-BN-Based Memristive Circuits: The Role of Each Type of Defect. *Advanced Materials*, 33(41), p.2103656.
4. Shi, Y., Liang, X., Yuan, B., Chen, V., Li, H., Hui, F., Yu, Z., Yuan, F., Pop, E., Wong, H.S.P. and Lanza, M., 2018. Electronic synapses made of layered two-dimensional materials. *Nature Electronics*, 1(8), pp.458-465.
5. Kund, M., Beitel, G., Pinnow, C.U., Rohr, T., Schumann, J., Symanczyk, R., Ufert, K.D. and Muller, G., 2005, December. Conductive bridging RAM (CBRAM): An emerging non-volatile memory technology scalable to sub 20nm. In *IEEE International Electron Devices Meeting, 2005. IEDM Technical Digest.* (pp. 754-757). IEEE.
6. Bernard, Y., Renard, V.T., Gonon, P. and Jousseume, V., 2011. Back-end-of-line compatible conductive bridging RAM based on Cu and SiO₂. *Microelectronic Engineering*, 88(5), pp.814-816.
7. Ielmini, D., 2011. Modeling the universal set/reset characteristics of bipolar RRAM by field-and temperature-driven filament growth. *IEEE Transactions on Electron Devices*, 58(12), pp.4309-4317
8. Ielmini, D., Nardi, F. and Cagli, C., 2011. Physical models of size-dependent nanofilament formation and rupture in NiO resistive switching memories. *Nanotechnology*, 22(25), p.254022.
9. Huang, P., Liu, X.Y., Chen, B., Li, H.T., Wang, Y.J., Deng, Y.X., Wei, K.L., Zeng, L., Gao, B., Du, G. and Zhang, X., 2013. A physics-based compact model of metal-oxide-based RRAM DC and AC operations. *IEEE transactions on electron devices*, 60(12), pp.4090-4097.
10. Yu, S. and Wong, H.S.P., 2011. Compact modeling of conducting-bridge random-access memory (CBRAM). *IEEE Transactions on Electron devices*, 58(5), pp.1352-1360
11. Choi, B.J., Torrezan, A.C., Norris, K.J., Miao, F., Strachan, J.P., Zhang, M.X., Ohlberg, D.A., Kobayashi, N.P., Yang, J.J. and Williams, R.S., 2013. Electrical performance and scalability of Pt dispersed SiO₂ nanometallic resistance switch. *Nano letters*, 13(7), pp.3213-3217.
12. Chen, Z., Huang, W., Zhao, W., Hou, C., Ma, C., Liu, C., Sun, H., Yin, Y. and Li, X., 2019. Ultrafast Multilevel Switching in Au/YIG/n-Si RRAM. *Advanced Electronic Materials*, 5(2), p.1800418.
13. Wang, C., Wu, H., Gao, B., Wu, W., Dai, L., Li, X. and Qian, H., 2017. Ultrafast RESET analysis of HfO_x-based RRAM by sub-nanosecond pulses. *Advanced Electronic Materials*, 3(12), p.1700263.
14. Shrestha, P.R., Nminibapiel, D., Kim, J.H., Campbell, J.P., Cheung, K.P., Deora, S., Bersuker, G. and Baumgart, H., 2014, June. Energy control paradigm for compliance-free reliable operation of RRAM. In *2014 IEEE International Reliability Physics Symposium* (pp. MY-10). IEEE.
15. Choi, B.J., Torrezan, A.C., Strachan, J.P., Kotula, P.G., Lohn, A.J., Marinella, M.J., Li, Z., Williams, R.S. and Yang, J.J., 2016. High-speed and low-energy nitride memristors. *Advanced Functional Materials*, 26(29), pp.5290-5296.
16. Lee, H.Y., Chen, Y.S., Chen, P.S., Gu, P.Y., Hsu, Y.Y., Wang, S.M., Liu, W.H., Tsai, C.H., Sheu, S.S., Chiang, P.C. and Lin, W.P., 2010, December. Evidence and solution of over-RESET problem for HfO_x based resistive memory with sub-ns switching speed and high endurance. In *2010 International Electron Devices Meeting* (pp. 19-7). IEEE.
17. Torrezan, A.C., Strachan, J.P., Medeiros-Ribeiro, G. and Williams, R.S., 2011. Sub-nanosecond switching of a tantalum oxide memristor. *Nanotechnology*, 22(48), p.485203.
18. Böttger, U., von Witzleben, M., Havel, V., Fleck, K., Rana, V., Waser, R. and Menzel, S., 2020. Picosecond multilevel resistive switching in tantalum oxide thin films. *Scientific reports*, 10(1), pp.1-9.

19. Zhang, F., Zhang, H., Shrestha, P.R., Zhu, Y., Maize, K., Krylyuk, S., Shakouri, A., Campbell, J.P., Cheung, K.P., Bendersky, L.A. and Davydov, A.V., 2018, December. An ultra-fast multi-level MoTe₂-based RRAM. In *2018 IEEE International Electron Devices Meeting (IEDM)* (pp. 22-7). IEEE.
20. Kim, M., Ducournau, G., Skrzypczak, S., Yang, S.J., Szriftgiser, P., Wainstein, N., Stem, K., Happy, H., Yalon, E., Pallecchi, E. and Akinwande, D., 2022. Monolayer molybdenum disulfide switches for 6G communication systems. *Nature Electronics*, pp.1-7.
21. Shi, Y., Liang, X., Yuan, B., Chen, V., Li, H., Hui, F., Yu, Z., Yuan, F., Pop, E., Wong, H.S.P. and Lanza, M., 2018. Electronic synapses made of layered two-dimensional materials. *Nature Electronics*, 1(8), pp.458-465.
22. Yang, Y., Gao, P., Li, L., Pan, X., Tappertzhofen, S., Choi, S., Waser, R., Valov, I. and Lu, W.D., 2014. Electrochemical dynamics of nanoscale metallic inclusions in dielectrics. *Nature communications*, 5(1), p.4232.
23. Yang, Y., Gao, P., Gaba, S., Chang, T., Pan, X. and Lu, W., 2012. Observation of conducting filament growth in nanoscale resistive memories. *Nature communications*, 3(1), p.732.
24. Dong, Z., Hua, Q., Xi, J., Shi, Y., Huang, T., Dai, X., Niu, J., Wang, B., Wang, Z.L. and Hu, W., 2023. Ultrafast and Low-Power 2D Bi₂O₂Se Memristors for Neuromorphic Computing Applications. *Nano Letters*, 23(9), pp.3842-3850.
25. Roldan, J.B., Maldonado, D., Aguilera-Pedregosa, C., Moreno, E., Aguirre, F., Romero-Zaliz, R., García-Vico, A.M., Shen, Y. and Lanza, M., 2022. Spiking neural networks based on two-dimensional materials. *npj 2D Materials and Applications*, 6(1), p.63.
26. Pan, C., Ji, Y., Xiao, N., Hui, F., Tang, K., Guo, Y., Xie, X., Puglisi, F.M., Larcher, L., Miranda, E. and Jiang, L., 2017. Coexistence of grain-boundaries-assisted bipolar and threshold resistive switching in multilayer hexagonal boron nitride. *Advanced functional materials*, 27(10), p.1604811

REVIEWER COMMENTS

Reviewer #1 (Remarks to the Author):

The authors have made a good revision and I don't have any other concern. I think the manuscript is publishable.

Reviewer #2 (Remarks to the Author):

Although the authors have made efforts to address the questions raised, however, the responses are less than satisfactory. Considering an insufficient advance in the device performance and a lack of clear understanding of the switching mechanism without solid evidence, I cannot support its publication in Nature Communications.

1. The ultra-large device-to-device variation in SET voltage (spanning 1V to 3V) is not practical for crossbar arrays. This will significantly reduce online learning accuracy and hinder the implementation of neural networks. Hence, the experimental demonstration reported here does not provide any technological advances to command the immediate interests of a broader audience.
2. The proposed switching mechanism failed to be fully supported by analytical TEM evidence of Ti filament, nor being detected using conductive AFM. This casts an ambiguity on the mechanism, and it should not be brushed off without a solid explanation. In the absence of clarity, this manuscript is falling short of the standard required for this journal.
3. The devices demonstrated in this work show an operating voltage much larger than 1V, which is not compelling for achieving low energy consumption. Again, this does not constitute a technological advance that warrant its publication in this journal.

Reviewer #3 (Remarks to the Author):

The authors have done an excellent job addressing each of my previous comments. The model and details explaining why 2D hBN has a higher switching speed than traditional TMO devices was particularly enlightening. I recommend this version for publication.

Reviewer #4 (Remarks to the Author):

There is a significant improvement in the revised manuscript. I am convinced to accept it for publication.

Ultra-fast switching memristors based on two-dimensional materials

Author response to reviewer comments

Firstly, we would like to thank all the reviewers for taking the time to review and provide valuable feedback to improve the quality of our manuscript. We have updated and modified the manuscript to reflect the suggestions. Please see below the detailed point-by-point response for each comment.

Reviewer 2

Although the authors have made efforts to address the questions raised, however, the responses are less than satisfactory. Considering an insufficient advance in the device performance and a lack of clear understanding of the switching mechanism without solid evidence, I cannot support its publication in Nature Communications.

Thank you for taking time to review the paper and provide useful comments that helped to improve the quality of the manuscript.

On the device performance front, our devices display superior performance in terms of switching speed and endurance at high frequencies, and on-par performance in terms of switching energy and retention time. We agree that the lower device-to-device variability would have been desirable, but we believe that this can be controlled and reduced using superior process control (as explained in Supplementary Figure.9). On the switching mechanism front, as suggested, we have conducted CAFM measurements and were able to clearly image the filaments. This new experimental result reinforces our proposed switching mechanism of Ti ion diffusion into the hBN layer. Please see detailed point by point explanation for all the comments below.

1. The ultra-large device-to-device variation in SET voltage (spanning 1V to 3V) is not practical for crossbar arrays. This will significantly reduce online learning accuracy and hinder the implementation of neural networks. Hence, the experimental demonstration reported here does not provide any technological advances to command the immediate interests of a broader audience.

Thank you for highlighting the concern about device-to-device variability. While we agree that the device-to-device variation is undesirable, we believe that this variation can be reduced by adopting superior processing methods (Supplementary Fig.9).

Upon closer examination of the 2D memristor literature, it becomes evident that only a small number of articles present statistical data on variability. Interestingly, all these articles utilize thicker switching layers (>5nm) in their memristors [1-5]. Thicker layers normalize the process variations induced by the metal surface roughness (order of ~1nm-2nm) or defects introduced by transfer process, resulting in reduced device-to-device variability. Even some memristors with thicker switching layers exhibit higher variability ($6\sigma > 2V$) [6-8]. On the other hand, for thinner 2D devices, device-to-device variability data is typically not disclosed [9-14]. In this article, we decided to report the device-to-device variability concern that needs the attention of the research community.

More importantly, the purpose of this article is to explore the limits of 2D hBN as a switching material from a switching speed perspective. Towards this end, we used thinner hBN layers for fabricating memristors that are known to be less tolerant to process variations. Had we utilized thicker hBN switching layer (>20 layers), our devices would have displayed reduced variability but at the cost of switching speeds. This trade-off between the superior properties of thinner 2D materials and variability needs to be carefully studied and optimized. Addressing this concern is beyond the scope of any single research publication. However, we acknowledge that the variability concern needs to be addressed for making this technology viable.

We strongly believe this work provides a significant contribution to the understanding of the 2D memristor operation in the high frequency regime. Below we summarize the major technological contributions of this work,

1. The devices presented in this work switch consistently with sub-nanosecond pulses. This shows the potential of these devices to be used for high-frequency applications.
2. The test setup and unique experimental methodology employed in this work allowed us to study the dynamic evolution of resistance (during the pulse) as well as the change in resistance (before and after the pulse). To the best of our knowledge, this is the first instance where such an approach has been adopted. Correlating these data points provided some key insights into the switching mechanism, which would otherwise remain undetected. For example, a significant amount of Joule-heating-induced filament narrowing was observed in our experiments.
3. The statistical analysis of transient switching characteristics presented in this work is novel and has never been reported. Statistical data unveiled some unique trends such as switching time-resistance ratio, switching energy-resistance ratio etc.
4. We present an electro-thermal model to capture the switching characteristics as well as the secondary effects (Joule heating). In addition, the fundamental reason for the ultra-fast switching nature of the 2D memristors has been explored with the help of finite element based physics solver.

Overall, we holistically studied the ultra-fast switching characteristics of the 2D memristors through experiments, modeling and statistical analysis. Through this work, we demonstrated that the 2D memristive devices are a potential avenue to explore for high frequency memory applications.

2. The proposed switching mechanism failed to be fully supported by analytical TEM evidence of Ti filament, nor being detected using conductive AFM. This casts an ambiguity on the mechanism, and it should not be brushed off without a solid explanation. In the absence of clarity, this manuscript is falling short of the standard required for this journal.

Thank you for your comment regarding the switching mechanism. As we mentioned in our earlier response report, the contrast difference between Ti and the background B,N is not sufficient to image the filament using TEM. Therefore, memristor studies using Ti/hBN system report defects/amorphous regions rather than filaments [8,15]. All the previous studies (2D materials or otherwise) that have successfully imaged the filament have Ag, Au or other heavy metal contacts that produce sufficient contrast for TEM imaging [1,16-18].

Although the filament cannot be observed, the electron energy loss spectroscopy (EELS) signal originating from the hBN film can provide an idea of the filament composition. Fig.2g (main text) presents the measured EELS data from the cross-section TEM profile. Evidently, the Ti peak can be observed within the hBN layers of a stressed memristor suggesting Ti ion diffusion into the switching layer.

Further, as suggested, to confirm the proposed filamentary conduction mechanism, we performed Conductive AFM (CAFM) measurements on our devices. Sample preparation process flow and experimental setup are shown in the schematic below (Figure R2.1). This approach of using ionic liquid for CAFM measurements was adopted from previous work [1].

Figure R2.1 Schematic showing the sample preparation process flow for Conductive AFM measurement. (a) Initial Si/SiO₂ substrate (SiO₂ thickness – 20nm) (b) Pattern and etch SiO₂ such that the Si surface is exposed (c) PVD deposition of Au followed by Ti (d) Transfer hBN film onto the bottom electrode (Ti comes in contact with hBN) (e) Spin coat PMMA layer (f) pattern and develop PMMA layer (small region 3x3um² exposed) (f) Drop Ionic liquid (makes contact with hBN layer) and apply voltage bias.

The sample was prepared such that the Ti metal made contact with the hBN layer to emulate our device structure. Filaments were formed in the hBN layer by stressing the device as shown in figure R2.1g (positive voltage to the bottom electrode). After stressing the devices, the ionic liquid was washed away, PMMA was dissolved in acetone and then the samples were taken to CAFM for measurements. During the CAFM measurements, the AFM tip comes in contact with the hBN layer and the back-side Si substrate is grounded. Large area scan (300nmx300nm) clearly shows the formation of conductive filaments (Figure R2.2a) across the hBN layer. High resolution scan of a single conducting filament is presented in Figure R2.2b.

Figure R2.2 (a) Large area CAFM current map showing multiple filaments (b) High resolution current map of a single filament.

Based on the CAFM and EELS measurements (Fig.2g, main text), we can confidently claim that the dominant switching mechanism in our devices originates from the formation/dissolution of Ti ion constituted conductive filaments.

3. The devices demonstrated in this work show an operating voltage much larger than 1V, which is not compelling for achieving low energy consumption. Again, this does not constitute a technological advance that warrant its publication in this journal.

Thank you for your comment.

The devices presented in this work have a mean DC SET voltage of 1.73V. Among these 80% of the data points are lower than 2V operating voltage. Most stable memristors (2D material and Transition metal oxide) have an operating voltage between 1V-2V [19-22]. Therefore, from a DC operating voltage perspective, our devices are on-par with reported literature.

When it comes to pulse testing, shorter pulses require higher amplitudes to deliver the energy required to overcome the barrier associated with filament formation/dissolution. Figure R2.3 shows the relationship between pulse amplitude and pulse width in our devices. Especially, the graph displays a steeper slope with ultra-short voltage pulses ($T_{PULSE} < 5ns$). This voltage-switching speed dependence can be effectively captured by an exponential function. This phenomenon of increasing pulse amplitude at higher frequencies has been reported by several studies [23-24,29].

Figure R2.3 Pulse amplitude variation with pulse width

		Material	Switching time	SET/REST voltage	Number of cycles presented	Reference
1	Transition Metal Oxide	Yttrium Iron Garnet	540ps	15V/-11V	20	23
2		HfO _x	100ps	-7.5V	1	24
3		HfO ₂	100ps	6V	1	25
4		AlN	85ps	1.9V/-2.1V	1	26
5		SiO ₂	105ps	5.5V/-4.5V	10	27
6		TaO _x	100ps	2.75V/-4.5V	1	28

7		TaO _x	250ps	9.21V	1	29
		MoTe ₂	5ns	1.25V	1	30
8	2D Material	MoS ₂	15ns	5V	1	31
9		MoS ₂	700ps	6V/-6V	1	32
10		MoS ₂	15ns	5V	1	33
11		MoTe ₂	100ns	1V/-2V	1	34
12		hBN (this work)	1.43ns	2.75V/2.25V	600	-
		hBN (this work)	120ps	4V/-4V	100	-

Table R2.1 Table showing the pulse amplitudes and pulse widths of different ultra-fast memristors

Table R2.1 shows the pulse amplitude, pulse widths and endurance cycling of various ultra-fast switching memristors. Although slower, we have also included other 2D memristors for comparison. Evidently, our devices operate at on par or lower voltages than most devices in the sub-nanosecond switching regime, with the exception of the device reported in [26]. The device [26] has a conducting channel in both ON/OFF states with different chemical compositions which leads to faster switching speeds at the cost of higher leakage in OFF state. Moreover, this report presents a single measurement collected at 100ps, and no statistical data was reported.. It is difficult to ascertain if the observed fast response is repeatable. For that matter, even our devices switch at lower operating voltages (2.65V at 120ps), but the switching is not consistent, as can be seen from Figure R2.4. The device could be successfully SET only on a few occasions (between cycles 20 to 25). Based on the criteria of a single switching cycle, we could also report a lower operating voltage (potentially lower than 1V), but we chose not to in view of the statistics..

Figure R2.4 Endurance cycling of our devices with lower voltage at 120ps.

References

1. Shi, Y., Liang, X., Yuan, B., Chen, V., Li, H., Hui, F., Yu, Z., Yuan, F., Pop, E., Wong, H.S.P. and Lanza, M., 2018. Electronic synapses made of layered two-dimensional materials. *Nature Electronics*, 1(8), pp.458-465.
2. Yuan, B., Liang, X., Zhong, L., Shi, Y., Palumbo, F., Chen, S., Hui, F., Jing, X., Villena, M.A., Jiang, L. and Lanza, M., 2020. 150 nm× 200 nm cross-point hexagonal boron nitride-based memristors. *Advanced Electronic Materials*, 6(12), p.1900115.
3. Yeh, C.H., Zhang, D., Cao, W. and Banerjee, K., 2020, December. 0.5 T0. 5R-Introducing an Ultra-Compact Memory Cell Enabled by Shared Graphene Edge-Contact and h-BN Insulator. In *2020 IEEE International Electron Devices Meeting (IEDM)* (pp. 12-3). IEEE.
4. Shen, Y., Zheng, W., Zhu, K., Xiao, Y., Wen, C., Liu, Y., Jing, X. and Lanza, M., 2021. Variability and Yield in h-BN-Based Memristive Circuits: The Role of Each Type of Defect. *Advanced Materials*, 33(41), p.2103656.
5. Zhuang, P., Lin, W., Ahn, J., Catalano, M., Chou, H., Roy, A., Quevedo-Lopez, M., Colombo, L., Cai, W. and Banerjee, S.K., 2020. Nonpolar resistive switching of multilayer-hBN-based memories. *Advanced Electronic Materials*, 6(1), p.1900979.
6. Krishnaprasad, A., Dev, D., Shawkat, M.S., Martinez-Martinez, R., Islam, M.M., Chung, H.S., Bae, T.S., Jung, Y. and Roy, T., 2023. Graphene/MoS₂/SiO_x memristive synapses for linear weight update. *npj 2D Materials and Applications*, 7(1), p.22.
7. Zhu, K., Liang, X., Yuan, B., Villena, M.A., Wen, C., Wang, T., Chen, S., Hui, F., Shi, Y. and Lanza, M., 2019. Graphene–boron nitride–graphene cross-point memristors with three stable resistive states. *ACS applied materials & interfaces*, 11(41), pp.37999-38005.
8. Roldan, J.B., Maldonado, D., Aguilera-Pedregosa, C., Moreno, E., Aguirre, F., Romero-Zaliz, R., García-Vico, A.M., Shen, Y. and Lanza, M., 2022. Spiking neural networks based on two-dimensional materials. *npj 2D Materials and Applications*, 6(1), p.63.
9. Kim, M., Ducournau, G., Skrzypczak, S., Yang, S.J., Szriftgiser, P., Wainstein, N., Stern, K., Happy, H., Yalon, E., Pallecchi, E. and Akinwande, D., 2022. Monolayer molybdenum disulfide switches for 6G communication systems. *Nature Electronics*, 5(6), pp.367-373.
10. Wu, X., Ge, R., Chen, P.A., Chou, H., Zhang, Z., Zhang, Y., Banerjee, S., Chiang, M.H., Lee, J.C. and Akinwande, D., 2019. Thinnest nonvolatile memory based on monolayer h-BN. *Advanced Materials*, 31(15), p.1806790.
11. Ge, R., Wu, X., Kim, M., Shi, J., Sonde, S., Tao, L., Zhang, Y., Lee, J.C. and Akinwande, D., 2018. Atomristor: nonvolatile resistance switching in atomic sheets of transition metal dichalcogenides. *Nano letters*, 18(1), pp.434-441.
12. Ge, R., Wu, X., Liang, L., Hus, S.M., Gu, Y., Okogbue, E., Chou, H., Shi, J., Zhang, Y., Banerjee, S.K. and Jung, Y., 2021. A library of atomically thin 2D materials featuring the conductive-point resistive switching phenomenon. *Advanced Materials*, 33(7), p.2007792.
13. Kim, M., Ge, R., Wu, X., Lan, X., Tice, J., Lee, J.C. and Akinwande, D., 2018. Zero-static power radio-frequency switches based on MoS₂ atomristors. *Nature communications*, 9(1), p.2524.
14. Kim, M., Pallecchi, E., Ge, R., Wu, X., Ducournau, G., Lee, J.C., Happy, H. and Akinwande, D., 2020. Analogue switches made from boron nitride monolayers for application in 5G and terahertz communication systems. *Nature Electronics*, 3(8), pp.479-485.
15. Pan, C., Ji, Y., Xiao, N., Hui, F., Tang, K., Guo, Y., Xie, X., Puglisi, F.M., Larcher, L., Miranda, E. and Jiang, L., 2017. Coexistence of grain-boundaries-assisted bipolar and threshold resistive switching in multilayer hexagonal boron nitride. *Advanced functional materials*, 27(10), p.1604811.
16. Yang, Y., Gao, P., Li, L., Pan, X., Tappertzhofen, S., Choi, S., Waser, R., Valov, I. and Lu, W.D., 2014. Electrochemical dynamics of nanoscale metallic inclusions in dielectrics. *Nature communications*, 5(1), p.4232.
17. Yang, Y., Gao, P., Gaba, S., Chang, T., Pan, X. and Lu, W., 2012. Observation of conducting filament growth in nanoscale resistive memories. *Nature communications*, 3(1), p.732.

18. Dong, Z., Hua, Q., Xi, J., Shi, Y., Huang, T., Dai, X., Niu, J., Wang, B., Wang, Z.L. and Hu, W., 2023. Ultrafast and Low-Power 2D Bi₂O₂Se Memristors for Neuromorphic Computing Applications. *Nano Letters*, 23(9), pp.3842-3850.
19. Wong, H.S.P., Lee, H.Y., Yu, S., Chen, Y.S., Wu, Y., Chen, P.S., Lee, B., Chen, F.T. and Tsai, M.J., 2012. Metal–oxide RRAM. *Proceedings of the IEEE*, 100(6), pp.1951-1970.
20. Shen, Z., Zhao, C., Qi, Y., Xu, W., Liu, Y., Mitrovic, I.Z., Yang, L. and Zhao, C., 2020. Advances of RRAM devices: Resistive switching mechanisms, materials and bionic synaptic application. *Nanomaterials*, 10(8), p.1437.
21. Chen, S., Mahmoodi, M.R., Shi, Y., Mahata, C., Yuan, B., Liang, X., Wen, C., Hui, F., Akinwande, D., Strukov, D.B. and Lanza, M., 2020. Wafer-scale integration of two-dimensional materials in high-density memristive crossbar arrays for artificial neural networks. *Nature Electronics*, 3(10), pp.638-645.
22. Yin, L., Cheng, R., Wen, Y., Liu, C. and He, J., 2021. Emerging 2D memory devices for in-memory computing. *Advanced Materials*, 33(29), p.2007081.
23. Chen, Z., Huang, W., Zhao, W., Hou, C., Ma, C., Liu, C., Sun, H., Yin, Y. and Li, X., 2019. Ultrafast Multilevel Switching in Au/YIG/n-Si RRAM. *Advanced Electronic Materials*, 5(2), p.1800418.
24. Wang, C., Wu, H., Gao, B., Wu, W., Dai, L., Li, X. and Qian, H., 2017. Ultrafast RESET analysis of HfO_x-based RRAM by sub-nanosecond pulses. *Advanced Electronic Materials*, 3(12), p.1700263.
25. Shrestha, P.R., Nminibapiel, D., Kim, J.H., Campbell, J.P., Cheung, K.P., Deora, S., Bersuker, G. and Baumgart, H., 2014, June. Energy control paradigm for compliance-free reliable operation of RRAM. In 2014 IEEE International Reliability Physics Symposium (pp. MY-10). IEEE.
26. Choi, B.J., Torrezan, A.C., Strachan, J.P., Kotula, P.G., Lohn, A.J., Marinella, M.J., Li, Z., Williams, R.S. and Yang, J.J., 2016. High-speed and low-energy nitride memristors. *Advanced Functional Materials*, 26(29), pp.5290-5296.
27. Choi, B.J., Torrezan, A.C., Norris, K.J., Miao, F., Strachan, J.P., Zhang, M.X., Ohlberg, D.A., Kobayashi, N.P., Yang, J.J. and Williams, R.S., 2013. Electrical performance and scalability of Pt dispersed SiO₂ nanometallic resistance switch. *Nano letters*, 13(7), pp.3213-3217.
28. Torrezan, A.C., Strachan, J.P., Medeiros-Ribeiro, G. and Williams, R.S., 2011. Sub-nanosecond switching of a tantalum oxide memristor. *Nanotechnology*, 22(48), p.485203.
29. Böttger, U., von Witzleben, M., Havel, V., Fleck, K., Rana, V., Waser, R. and Menzel, S., 2020. Picosecond multilevel resistive switching in tantalum oxide thin films. *Scientific reports*, 10(1), pp.1-9.
30. Zhang, F., Zhang, H., Shrestha, P.R., Zhu, Y., Maize, K., Krylyuk, S., Shakouri, A., Campbell, J.P., Cheung, K.P., Bendersky, L.A. and Davydov, A.V., 2018, December. An ultra-fast multi-level MoTe₂-based RRAM. In 2018 IEEE International Electron Devices Meeting (IEDM) (pp. 22-7). IEEE.
31. Ge, R., Wu, X., Kim, M., Chen, P.A., Shi, J., Choi, J., Li, X., Zhang, Y., Chiang, M.H., Lee, J.C. and Akinwande, D., 2018, December. Atomristors: Memory effect in atomically-thin sheets and record RF switches. In 2018 IEEE International Electron Devices Meeting (IEDM) (pp. 22-6). IEEE.
32. Kim, M., Ducournau, G., Skrzypczak, S., Yang, S.J., Szriftgiser, P., Wainstein, N., Stern, K., Happy, H., Yalon, E., Pallecchi, E. and Akinwande, D., 2022. Monolayer molybdenum disulfide switches for 6G communication systems. *Nature Electronics*, pp.1-7.
33. Wu, X., Ge, R., Kim, M., Akinwande, D. and Lee, J.C., 2020, February. Atomristors: Non-volatile resistance switching in 2D monolayers. In 2020 Pan Pacific Microelectronics Symposium (Pan Pacific) (pp. 1-6). IEEE.

34. Zhang, F., Zhang, H., Krylyuk, S., Milligan, C.A., Zhu, Y., Zemlyanov, D.Y., Bendersky, L.A., Burton, B.P., Davydov, A.V. and Appenzeller, J., 2019. Electric-field induced structural transition in vertical MoTe₂-and Mo_{1-x}W_xTe₂-based resistive memories. *Nature materials*, 18(1), pp.55-61.